# OpInf-LLM: Parametric PDE Solving with LLMs via Operator Inference

## Abstract

Solving diverse partial differential equations (PDEs) is fundamental in science and engineering. Large language models (LLMs) have demonstrated strong capabilities in code generation, symbolic reasoning, and tool use, but reliably solving PDEs across heterogeneous settings remains challenging. Prior work on LLM-based code generation and transformer-based foundation models for PDE learning has shown promising advances. However, a persistent trade-off between execution success rate and numerical accuracy arises, particularly when generalization to unseen parameters and boundary conditions is required. In this work, we propose OpInf-LLM, an LLM parametric PDE solving framework via operator inference. The proposed framework leverages small amounts of solution data to enable accurate prediction of diverse PDE instances, including unseen parameters and configurations, and provides seamless integration with LLMs for natural language task specification and physics-based reasoning of proper feature parameterization. Its low computational demands and unified solution pipeline further enable a high execution success rate across heterogeneous settings, opening new possibilities for generalizable reduced-order modeling in LLM-based PDE solving.

## 1 Introduction

Solving diverse partial differential equations (PDEs) is fundamental in science and engineering, underpinning applications ranging from fluid dynamics and heat transfer to electromagnetics, materials science, and climate modeling (Cai et al., 2021; Oommen et al., 2024; Nohra & Dufour, 2024; Pathak et al., 2022). In practice, modern PDE workflows must accommodate heterogeneous governing equations, parameter regimes, and boundary and initial conditions, often under strict accuracy and robustness requirements. As a result, there is growing interest in automated and generalizable PDE solvers that can flexibly adapt to new problem instances without extensive manual intervention or retraining (Karniadakis et al., 2021; Li et al., 2020; Lu et al., 2019; Sun et al., 2025; Ye et al., 2024; Liu et al., 2024a).

Large language models (LLMs) have recently demonstrated strong capabilities in code generation (Jiang et al., 2024; Huynh & Lin, 2025), symbolic reasoning (Xu et al., 2024; Shojaee et al., 2024), and tool use (Xu et al., 2025b), motivating a new line of work on language-driven scientific computing (Nejjar et al., 2025). In the context of PDEs, prior work on LLM-based code generation and transformer-based foundation models for PDE learning has shown promising advances, including automatic solver synthesis and data-driven surrogate modeling (Li et al., 2025; Gaonkar et al., 2025; Wu et al., 2025; Negrini et al., 2025). However, reliably solving PDEs across heterogeneous settings remains challenging for LLM-based approaches. In particular, a persistent trade-off between execution success rate and numerical accuracy arises, especially when generalization to unseen parameters, boundary conditions, and domain configurations is required (Fig. 1). Code-generation-based methods often suffer from brittle execution and solver instability, while purely data-driven neural surrogates can exhibit poor extrapolation and high training costs.

These limitations highlight a fundamental gap between expressive, flexible language-driven interfaces and robust, generalizable numerical solvers. On the one hand, LLMs provide a powerful abstraction layer for specifying PDEs and solver configurations via natural language (Li et al., 2024a). On the other hand, classical

reduced-order modeling techniques, such as operator inference, offer principled mechanisms for constructing compact, interpretable, and generalizable dynamical system representations from limited data, but cannot handle diverse PDEs at once with language specifications (Peherstorfer & Willcox, 2016; Kramer et al., 2024). Bridging these two paradigms offers a promising pathway toward reliable, language-driven PDE solving.

In this work, we propose OpInf-LLM, an LLM parametric PDE solving framework via operator inference (Fig. 2). We first leverage operator inference to learn reduced-order models for diverse parametric PDE instances from a small amount of solution data over a finite set of parameter values and configuration settings, yielding a shared reduced basis and parameter-dependent reduced operators. We then integrate large language models to infer and solve new reduced-order models via agentic tool calling or direct code generation, enabling the prediction of PDE solutions for diverse instances, including previously unseen parameters and varying boundary conditions. By construction, the reduced-order model effectively reduces the complexity of PDE generalization to polynomial fitting over learned operators and time integration of a low-dimensional ODE system, resulting in low computational demands at test time and a high execution success rate across heterogeneous PDE settings. Moreover, the use of LLMs naturally admits natural language instructions for specifying PDEs and configurations, while reasoning according to the underlying physics for proper feature identification, thereby providing a flexible and unified framework for diverse PDE-solving tasks. The key advantages of OpInf-LLM are:

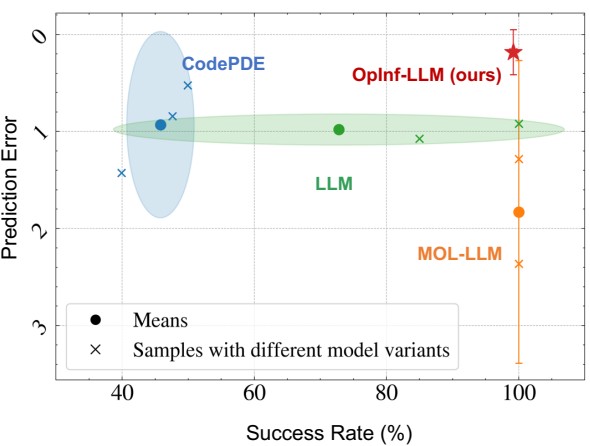

Figure 1: Trade-offs in prediction error and solver execution success rate. The mean and variance are computed from the experimental results in Section 4 across the three benchmark problems considered in this study.

- Accurate prediction of PDE solutions under diverse unseen parameters and configuration settings.

- Lightweight training with high execution success rates across heterogeneous PDEs and settings in one unified framework.

- Natural-language problem specification and physically consistent identification of operator features for parametric PDE solving.

## 2 Related Work

**Operator Inference.** Reduced-order modeling (ROM) (Benner et al., 2015; Brunton & Kutz, 2022) is a class of model reduction techniques that aims to reduce the computational cost of simulating large-scale dynamical systems governed by PDEs by leveraging a low-dimensional latent space. Among model reduction techniques, operator inference (OpInf) (Kramer et al., 2024; Peherstorfer & Willcox, 2016) is a data-driven ROM method that learns structured reduced dynamics consistent with PDE nonlinearities, allowing accurate prediction with improved generalization and lower data requirements compared to black-box approaches. OpInf has been applied to parametric settings (McQuarrie et al., 2023), used to simulate complex physics such as combustion chamber dynamics (McQuarrie et al., 2021; Swischuk et al., 2020), and enhancements have been proposed to account for a wider class of PDE nonlinearities (Qian et al., 2020; 2022) or enhance robustness of the predictions (Sawant et al., 2023). However, existing OpInf methods generally require the reduced model structure, regression features, data-processing steps, and solver implementation to be manually specified for each PDE family. To the best of our knowledge, ROMs have not been explored for language-model-driven PDE solving or for extending beyond single parametric PDE families. We address this gap by bridging OpInf with expressive LLM reasoning and coding capabilities, enabling the automatic construction and implementation of reduced-order solvers from natural-language PDE descriptions across diverse problem families.

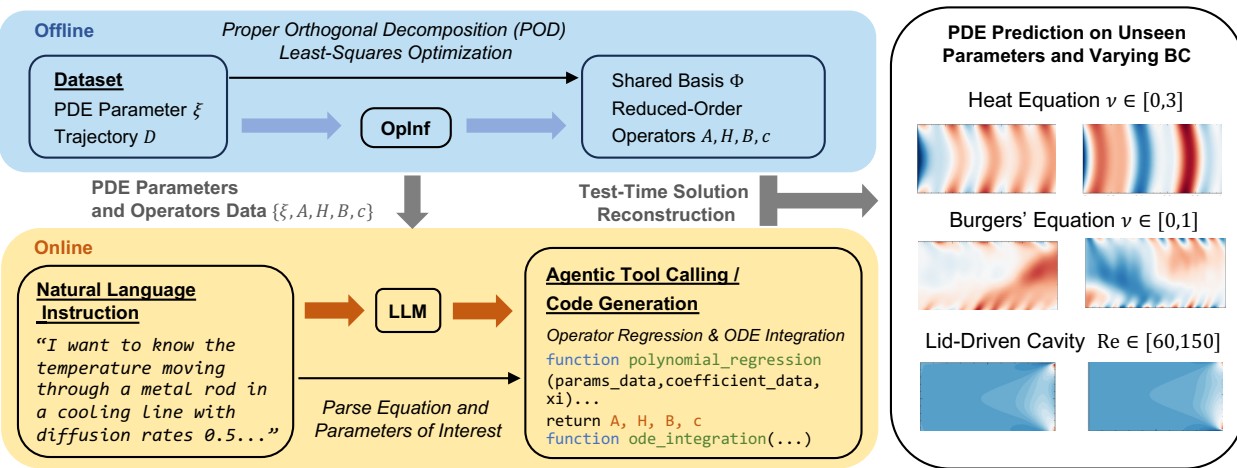

Figure 2: Overall diagram of the OpInf-LLM framework. In the offline stage, shared basis and reduced-order operators are obtained from limited amounts of PDE solution data via proper orthogonal decomposition (POD) and least-squares optimization. In the online stage, LLM parses natural language instructions, generates code or calls agentic tools for operator regression and reduced-order ODE integration, to predict PDE solutions of diverse parametric and boundary conditions, including configurations beyond the available dataset.

**PDE Foundation Models.** Recent advances in AI/ML have led to the development of a broad class of data-driven PDE solvers, including physics-informed neural networks (PINNs) (Raissi et al., 2019; Han et al., 2018; Cuomo et al., 2022; Cho et al., 2024; Huang et al., 2022; Wang et al., 2025b), neural operators (Kovachki et al., 2023; Li et al., 2020; Lu et al., 2019; 2021; 2022; Li et al., 2024b; Goswami et al., 2023; Wang et al., 2025a), and neural PDE solvers (Brandstetter et al., 2022; Takamoto et al., 2023; Hsieh et al., 2019). More recently, PDE foundation models that learn generalizable representations across diverse equation families have been proposed (Herde et al., 2024; Liu et al., 2024b; Ye et al., 2024; Sun et al., 2025; Shen et al., 2024; Rautela et al., 2025; Zhu et al., 2025; Zhou et al., 2025; Hao et al., 2024; Wiesner et al., 2025). These models aim to solve multiple PDEs with a single pre-trained architecture, reducing the need for equation-specific training. Building on this direction, multimodal approaches have emerged that incorporate language inputs to enhance flexibility and interpretability (Negrini et al., 2025; Bao et al., 2025). In parallel, diffusion-based neural PDE solvers have been developed to support text-conditioned simulation using captions generated by multimodal LLMs (Zhou et al., 2024). Additionally, LLMs have been explored as auxiliary modalities to improve surrogate model performance (Lorsung & Farimani, 2024). Despite this progress, robust generalization to unseen parameters and initial and boundary condition settings remains a significant challenge.

**LLMs for Scientific Machine Learning.** Large language models have shown promising capabilities in scientific machine learning. Prior works focus on using LLMs to directly generate executable PDE solvers from natural language descriptions (Li et al., 2025; Wu et al., 2025), as well as leveraging LLMs to write code for broader scientific machine learning problems (Gaonkar et al., 2025; Jiang et al., 2025; He et al., 2025). Beyond code generation, Yang et al. (2025) fine-tunes LLMs for in-context operator learning, while Xu et al. (2025a) explores PDE discovery by training generative models with textbook information. LLM-based agents have also adopted agentic workflows for PDE solving (Jiang & Karniadakis, 2025; Liu et al., 2025). In parallel, many efforts have explored LLM-driven autoformalization of PDEs, translating language descriptions into formal mathematical representations (Soroco et al., 2025; Mensfelt et al., 2025), building on broader autoformalization research that maps informal mathematical content into formal, machine-verifiable languages (Wu et al., 2022; Weng et al., 2025). Equation discovery from data has also been explored in Du et al. (2024); Grayeli et al. (2024). Despite these advances, solver reliability and execution robustness remain key challenges.

## 3 Proposed Method

In this section, we present the proposed OpInf-LLM method. The overall diagram is shown in Fig. 2. The method consists of two stages. In the offline stage, operator inference (OpInf) is used to construct a parametric reduced-order model (ROM) by learning a projection basis and reduced modal dynamics from trajectory data of PDE systems with finite parameter coverage. In the online stage, a large language model (LLM) is integrated at test time to enable the prediction of solutions for diverse PDE instances based on natural-language instructions, while generalizing to previously unseen parameter configurations.

### 3.1 Operator Inference

We start by introducing OpInf, which is a data-driven model reduction method for PDEs (Kramer et al., 2024; Peherstorfer & Willcox, 2016) and is the key component in the offline stage of the proposed method.

Consider a general class of nonlinear parametric PDEs:

$$\frac{\partial y}{\partial t} = \mathcal{F}(\frac{\partial y}{\partial x}, \frac{\partial^2 y}{\partial x^2}, \cdots, s, \xi), \quad x \in \Omega, \quad t \in [0, T], \tag{1}$$

with initial and boundary conditions

$$\begin{aligned} \mathcal{I}[y](x, 0) &= g(x), \quad x \in \Omega, \\ \mathcal{B}[y](x, t) &= u(t), \quad x \in \partial\Omega, \end{aligned} \tag{2}$$

where $\Omega \subset \mathbb{R}^d$ is the spatial domain, $s$ is a known source term, $\xi \in \mathbb{R}$ is the PDE parameter, and $\mathcal{F}$ is the generic nonlinear PDE operator.

OpInf learns a ROM approximating the PDE dynamics within a finite $r$-dimensional subspace spanned by a set of basis functions $\phi_1(x), \ldots, \phi_r(x)$. Specifically, the PDE state is decomposed as

$$y(x, t, \xi) = \sum_{i=1}^{r} a_i(t, \xi)\phi_i(x), \tag{3}$$

where $a_i(t, \xi)$ are time-dependent modal coefficients. In classical model reduction, the dynamics for these modal coefficients can be derived from the PDE itself using Galerkin projection (Holmes et al., 2012). For PDE operators $\mathcal{F}$ with at most quadratic nonlinearities, which encompass a broad class of PDEs commonly arising in physics (e.g., the diffusion equation modeling heat transfer or the Navier-Stokes equations modeling fluid dynamics), the modal coefficients inherit reduced dynamics of the following form given $\xi$:

$$\frac{da}{dt} = A(\xi)a + H(\xi)(a \otimes a) + B(\xi)u + c(\xi), \tag{4}$$

where $A \in \mathbb{R}^{r \times r}$ is a linear operator, $H \in \mathbb{R}^{r \times r^2}$ is a quadratic operator, $B \in \mathbb{R}^{r \times m}$ is a control input matrix, and $c \in \mathbb{R}^r$ is an offset vector, with $\otimes$ denoting the Kronecker product. This procedure naturally applies to PDE operators $\mathcal{F}$ with higher-order or non-polynomial nonlinearities, leading to modal dynamics with structures that differ from equation 4, for example through the inclusion of cubic terms (Qian et al., 2022). In this paper, we stick to quadratically nonlinear PDEs with the formulation in equation 4 for clarity.

For a fixed $\xi$, OpInf composes the following two steps: 1) identify an appropriate basis $\phi_1(x), \ldots, \phi_r(x)$; 2) identify the corresponding reduced-order operators $A$, $H$, $B$, $c$. These procedures yield a reduced-order model defined by equation 3 and equation 4 for solving PDEs with varying boundary conditions. For the rest of this subsection, we consider a single fixed $\xi$ and we omit $\xi$ dependency in the notation for conciseness.

**Basis identification.** We collect a dataset $\mathcal{D}$, obtained either from physical experiments or numerical simulations of the PDE in equation 1. The dataset consists of time series of the solution and its temporal derivatives over a discretization $\boldsymbol{x}$ of the full spatial domain

$$\mathcal{D} = \{y(\boldsymbol{x}, t_1), \ldots, y(\boldsymbol{x}, t_K)\} \cup \left\{ \frac{\partial y}{\partial t}(\boldsymbol{x}, t_1), \ldots, \frac{\partial y}{\partial t}(\boldsymbol{x}, t_K) \right\}. \tag{5}$$

For notational simplicity, equation 5 is written for a single trajectory. In practice, the dataset may include multiple trajectories corresponding to different initial conditions and boundary conditions. Then, for a given reduced-order basis dimension $r$, OpInf computes the basis functions by conducting a proper orthogonal decomposition (POD) (Chatterjee, 2000) of the concatenated solution values $y$ contained in the dataset $\mathcal{D}$, which yields an optimal basis that maximizes the captured energy. Specifically, $\Phi := [\phi_1(\boldsymbol{x}), \ldots, \phi_r(\boldsymbol{x})]$ are the first $r$ left-singular vectors of the trajectory data matrix $\{y(\boldsymbol{x}, t_1), \ldots, y(\boldsymbol{x}, t_K)\}$, corresponding to the $r$ largest singular values.[1] The number of basis functions $r$ can be tuned based on the desired level of captured energy (Peherstorfer & Willcox, 2016). Ablations on the effects of the basis function number $r$ can be found in Appendix C.2.

**Operator identification.** Once the POD basis has been obtained, we project the dataset $\mathcal{D}$ onto the basis functions to form a reduced-order dataset $\mathcal{A}$ of modal coefficients and their time derivatives. Specifically,

$$\mathcal{A} = \{a(t_1), \ldots, a(t_K)\} \cup \{\dot{a}(t_1), \ldots, \dot{a}(t_K)\}, \tag{6}$$

where

$$
\begin{aligned}
a_i(t_j) &= \langle y(\boldsymbol{x}, t_j), \phi_i(\boldsymbol{x}) \rangle, \\
\dot{a}_i(t_j) &= \left\langle \frac{\partial y}{\partial t}(\boldsymbol{x}, t_j), \phi_i(\boldsymbol{x}) \right\rangle,
\end{aligned}
\tag{7}
$$

for $i = 1, \ldots, r$ and $j = 1, \ldots, K$. Here $\langle \cdot, \cdot \rangle$ denotes the $L_2$ inner product. Then, OpInf calculates the operators $A, H, B$ and $c$ such that the reduced dynamics in equation 4 best fits the data in the least-squares sense, which can be formulated as the least-squares problem

$$\min_{A,H,B,c} \sum_{k=1}^{K} \|[Aa + H(a \otimes a) + Bu + c - \dot{a}](t_k)\|_F^2 + \lambda \left(\|A\|_F^2 + \|H\|_F^2 + \|B\|_F^2 + \|c\|_2^2\right), \tag{8}$$

where $\|\cdot\|_F$ denotes the Frobenius norm, and $\lambda$ is a Tikhonov regularization coefficient that prevents model overfitting and is often selected through a line search (Swischuk et al., 2020). Ablations on the effects of the regularization intensity $\lambda$ can be found in Appendix C.3.

During online inference, given any initial condition and boundary conditions of interest, the ROM approximates the PDE solution by first calculating the initial modal coefficients corresponding to the initial condition, then propagating the modal coefficients over time according to the reduced dynamics in equation 4, and reconstructing the full-order solution via equation 3.

### 3.2 Parametric OpInf and LLM Integration

In the previous section, we discussed how OpInf can learn ROMs for PDEs with a single fixed parameter $\xi$. In this section, we will discuss how OpInf can be extended to parametric PDEs, and how we integrate it with LLMs to solve diverse equations with natural language instructions.

The key insight behind parametric OpInf is as follows (McQuarrie et al., 2023). First, the ROM can share a common basis $\Phi$ within a parametric PDE family. Second, once reduced dynamics operators $A$, $H$, $B$, $c$ have been separately obtained for a set of training parameters $\xi_1, \xi_2, \ldots$, reduced dynamics operators for any unseen $\xi$ can be obtained via polynomial regression among the existing operators. Based on those insights, the proposed OpInf-LLM performs the following procedures to solve diverse parametric PDEs.

**Basis identification.** We collect a trajectory dataset $\mathcal{D}$ in the form of time series of solutions and time derivatives similar to equation 5, but for multiple parameters $\xi$

$$
\begin{aligned}
\mathcal{D} = \{&y(\boldsymbol{x}, t_1, \xi_1), \ldots, y(\boldsymbol{x}, t_K, \xi_1)\} \cup \{y(\boldsymbol{x}, t_1, \xi_2), \ldots\} \\
&\cdots \cup \left\{\frac{\partial y}{\partial t}(\boldsymbol{x}, t_1, \xi_1), \ldots\right\} \cup \left\{\frac{\partial y}{\partial t}(\boldsymbol{x}, t_1, \xi_2), \ldots\right\} \cdots.
\end{aligned}
\tag{9}
$$

---

[1]For PDEs with $d > 1$ spatial dimensions, solution snapshots $y(x, t)$ in the trajectory data matrix are vectorized (flattened) before performing the POD.

Here, equation 9 shows a single trajectory for each $\xi_i$ for notation conciseness, but multiple trajectories can be used. Then, for a given number of basis functions $r$, the shared basis $\Phi := [\phi_1(\boldsymbol{x}), \ldots, \phi_r(\boldsymbol{x})]$ can be obtained via proper orthogonal decomposition (POD) of all solution data $y$ in equation 9.

**Operator identification.** Once the POD basis is determined, the reduced-order dataset $\mathcal{A}$ of modal coefficients and their time derivatives can be obtained by projecting equation 7 on equation 9

$$
\begin{aligned}
\mathcal{A} = \{a(t_1, \xi_1), \ldots, a(t_K, \xi_1)\} \cup \{a(t_1, \xi_2), \ldots\} \cdots \\
\cup \{\dot{a}(t_1, \xi_1), \ldots, \dot{a}(t_K, \xi_1)\} \cup \{\dot{a}(t_1, \xi_2), \ldots\} \cdots .
\end{aligned}
\tag{10}
$$

Then for each $\xi_i$ in the dataset $\mathcal{A}$, we solve the least-squares problem in equation 8 to obtain the reduced operators $A(\xi_i)$, $H(\xi_i)$, $B(\xi_i)$, $c(\xi_i)$.

**OpInf-LLM.** Once the above procedures are executed, we integrate large language models (LLMs) to admit natural language instructions specifying the target PDE and its configurations, and leverage LLM tool-calling or code generation capabilities to infer the reduced operators and perform time integration, enabling parametric PDE solving across diverse settings. Specifically, given a natural language description of a PDE-solving task (e.g., "solve the viscous Burgers' equation with viscosity $\nu = 0.03...$"), the LLM first parses the governing equation and the specified parameters. Notably, the LLM can handle descriptive, unstructured, and non-technical instructions, as demonstrated by the ablations in Section 4.4. Based on this information, the LLM generates code or calls existing tools to perform operator regression and reduced-order ODE integration. Specifically, it performs interpolation or regression[2] on operator data to infer the parameter-dependent reduced operators $A(\xi)$, $H(\xi)$, $B(\xi)$, and $c(\xi)$ for the parameters $\xi$ of interest. Note that during this step the LLM also reasons about the underlying physics in choosing the parametric regression structure for these operators (see Section 4.4 for ablations and discussion). It then integrates the reduced-order ODE system in equation 4 using the parsed initial and boundary condition specifications to predict the PDE solution. The key features of this integration include: **(i).** Natural language descriptions of diverse PDEs as instructions; **(ii).** Low execution failure rate enabled by moderate interpolation/regression and reduced-order ODE integration demands; **(iii).** Accurate solution for unseen PDE parameters and ICBC configurations from a unified ROM structure.

## 4 Experiments

In this section, we evaluate the performance of the proposed OpInf-LLM framework and compare it against representative baseline methods. While many widely used PDE benchmarks focus on fixed or periodic boundary conditions (Li et al., 2020; Raissi et al., 2019; Takamoto et al., 2022; Gupta & Brandstetter, 2022), we consider parametric PDEs with varying non-homogeneous boundary conditions to highlight the generalization capabilities of the proposed approach. Full experimental details are provided in Appendix A.

### 4.1 PDEs of Interest

**Heat Equation.** We consider the one-dimensional heat equation

$$
y_t(x, t) = \nu\, y_{xx}(x, t), \qquad x \in [0, 1], \tag{11}
$$

subject to Dirichlet boundary conditions

$$
y(0, t) = y(1, t) = u(t), \tag{12}
$$

and a fixed smooth initial condition $y(x, 0) = g(x)$. The boundary input $u(t)$ is chosen as a multi-sine signal

$$
u(t) = b + \sum_{k=1}^{3} A_k \sin(2\pi f_k t + \phi_k), \tag{13}
$$

---

[2]In practice, both interpolation and regression are valid, and the proposed framework supports both options (see Appendix C.7).

where the amplitudes $A_k$, frequencies $f_k$, phases $\phi_k$, and bias $b$ are independently randomized for each trajectory. This construction provides persistent excitation across a range of temporal modes.

Training data are generated over a time horizon $T = 1.0$ using diffusivity values $\nu \in \{0.1, 0.5, 2.0\}$ with 3 trajectories per parameter. Number of POD modes is set to $r = 6$ for OpInf and regularization $\lambda = 10^{-6}$. Evaluation is performed on 20 trajectories each corresponding to diffusivity values $\nu \in \{0.5, 1.0, 3.0\}$.

**Burgers' Equation.** We consider the forced viscous Burgers' equation

$$y_t(x,t) + y\, y_x(x,t) = \nu\, y_{xx}(x,t) + s(x,t), \quad x \in [0,1], \tag{14}$$

with Dirichlet boundary conditions

$$y(0,t) = u_1(t), \qquad y(1,t) = u_2(t). \tag{15}$$

The boundary inputs $u_1(t)$ and $u_2(t)$ represent time-dependent boundary conditions, while $s(x,t)$ denotes an internal source term. Both the boundary inputs and the source term are constructed using multi-sine signals, with a fixed spatial profile used to localize the source term. The initial condition is chosen to be smooth and consistent with the boundary values at $t = 0$.

Training data are generated over a time horizon $T = 2.0$ using viscosity parameters $\nu \in \{0.01, 0.02, 0.05, 0.1\}$ with 100 trajectories per parameter. Number of POD modes is set to $r = 10$ for OpInf and regularization $\lambda = 0.5$. Evaluation is performed on 20 trajectories each corresponding to previously unseen viscosity values $\nu \in \{0.03, 0.07\}$, while maintaining the same boundary and forcing input structure.

**2D Lid-Driven Cavity Flow.** We consider two-dimensional incompressible flow in a square cavity using the vorticity-stream function formulation

$$\omega_t + v_1\, \omega_{p_1} + v_2\, \omega_{p_2} = \frac{1}{\mathrm{Re}}\left(\omega_{p_1 p_1} + \omega_{p_2 p_2}\right), \tag{16}$$

$$\Delta \psi = -\omega, \tag{17}$$

where $\omega$ is the vorticity, $x = (p_1, p_2) \in [0,1]^2$ denotes the two-dimensional spatial coordinate, subscripts $p_1$ and $p_2$ indicate corresponding partial derivatives, and the velocity $v = (v_1, v_2)$ is related to the stream function $\psi$ as

$$v_1 = \psi_{p_2}, \qquad v_2 = -\psi_{p_1}. \tag{18}$$

No-slip boundary conditions are enforced on all cavity walls. The top lid is driven by a spatially varying and time-dependent horizontal velocity

$$v_1(x,t) = h(p_1)\, u(t), \qquad v_2 = 0, \tag{19}$$

where $h(p_1)$ is a fixed, non-symmetric spatial profile, and $u(t)$ is a randomized sinusoidal input.

Training data are generated over a time horizon $T = 2.0$ using Reynolds numbers $\mathrm{Re} \in \{50, 75, 100, 125, 150\}$ with 8 trajectories per parameter. The number of POD modes is set to $r = 20$ for OpInf and the regularization strength is fixed to $\lambda = 3$. Model evaluation is carried out on 2 trajectories for each unseen Reynolds number $\mathrm{Re} \in \{60, 80, 90, 110, 120, 140\}$.

Across the three problems of interest, the training set contains 449 trajectories in total, consisting of 9 heat-equation trajectories, 400 Burgers' trajectories, and 40 lid-driven cavity trajectories. Evaluation is performed on 112 test trajectories in total, consisting of 60 heat-equation trajectories, 40 Burgers' trajectories, and 12 cavity-flow trajectories. Unless otherwise specified, the overall code execution success rate reported in Table 1 is computed over all 112 test trajectories.

## 4.2 Baselines

**CodePDE** (Li et al., 2025). CodePDE represents a class of LLM-based approaches that directly generate executable PDE solvers in a programming language from a textual description of the governing equations,

initial conditions, and boundary conditions. It employs a self-debugging mechanism that iteratively refines generated code using execution feedback. We apply test-time scaling with 5 independent solver generation rounds, each allowing 5 debugging trials, and report the best-performing solver.

**MOL-LLM** (Negrini et al., 2025). MOL-LLM represents multimodal PDE foundation models with language integration, combining a transformer-based PDE foundation model with language inputs. The model takes as input the initial state of the discretized solution, the PDE parameter vector, and boundary condition parameters, together with textual tokens describing the governing equation. Given these inputs, MOL-LLM outputs the full solution trajectory over a fixed time grid. The model is trained for 25000 steps as in the original implementation.

**MOL-LLM (large dataset).** Following the original MOL-LLM work, which is trained on large-scale datasets, we also evaluate a large-data variant of this baseline. In this setting, the same MOL-LLM model and training procedure are used, but the training set is expanded to 5000 trajectories per equation sampled from the same underlying distribution.

**LLMs.** We directly prompt large language models with the governing PDE, full initial and boundary conditions, external inputs, and a coarse output grid for tractability (heat/Burgers: 16 spatial points $\times$ 41 time steps; cavity: $6 \times 6$ spatial points $\times$ 21 time steps), and ask it to return the full spatiotemporal state array in JSON format (3-decimal precision) for a single parameter setting and trajectory. The LLM is treated as a black-box simulator, and its predicted field is compared against downsampled ground truth data using the relative $L_2$ error. For each parameter instance and equation, the LLM is queried 10 times, and the averaged error over all successful cases is reported. Example prompts are provided in Appendix B.3.

Note that these baselines differ substantially in their data requirements. CodePDE and direct LLM prediction are zero-training-data methods, since they rely only on the problem description at inference time. MOL-LLM, on the other hand, is a supervised foundation-model baseline that typically requires a large amount of trajectory data for training. The proposed OpInf-LLM framework lies between these two regimes, using only a small number of trajectories to learn a reduced-order model while still accepting natural-language problem descriptions. Although OpInf-LLM learns the dynamics in a reduced-order coordinate space, its predictions are reconstructed to the full physical solution field before evaluation, so the reported errors are computed in the same output space as other baselines. In this sense, OpInf-LLM fills an intermediate gap between purely code-generating or direct-prediction LLM approaches and data-intensive PDE foundation models, aiming to retain the flexibility of language-guided methods while achieving improved reliability and comparably low prediction error.

All experiments were conducted on a workstation with an Intel Core i7-14700KF CPU, 125 GiB RAM, and an NVIDIA GeForce RTX 4090 GPU. OpInf-LLM training time denotes the offline computation time for POD basis construction and solution projection.

### 4.3   Results

We report the results of the proposed OpInf-LLM framework and compare them with baseline methods. Table 1 summarizes the number of training samples, training time, code success rate, and the average relative $L_2$ error over $[0, T]$ for the heat, Burgers', and lid-driven cavity flow equations. Visualizations of prediction results across equation parameters and methods are shown in Fig. 3–5.

It can be seen that CodePDE does not consistently produce high-quality solvers, often resulting in either code failures or inaccurate numerical solutions, which limits its reliability in practice. MOL-LLM achieves a high execution success rate since it is primarily a transformer-based method, but its prediction error becomes large when generalizing to different parameters and boundary conditions is required, and its training cost is substantially higher than that of the other methods. Direct LLM prediction of PDE solutions remains challenging, as existing LLMs tend to infer boundary conditions correctly but zero out or purely interpolate the interior values, and their success rate is further reduced by shape or format mismatches. In contrast, OpInf-LLM demonstrates reliable overall performance across the tested equations, maintaining a high code execution success rate while achieving competitive prediction errors with substantially less training data than MOL-LLM. It accepts natural-language instructions and predicts diverse PDE solutions with varying

Table 1: Results summary. Average relative $L_2$ error is reported for each equation. **Bold** denotes the best result, and underline denotes the second best. ↑ higher is better, ↓ lower is better.

| Method | Data | Train time (s) | Success rate ↑ | Heat ↓ | Burgers ↓ | Cavity ↓ |
|---|---|---|---|---|---|---|
| CodePDE (GPT-4.1) | – | – | 49.9% | 1.59e−2 | **1.50e−1** | 1.41e0 |
| CodePDE (GPT-4o) | – | – | 39.9% | 1.60e−1 | 1.23e0 | 2.89e0 |
| CodePDE (Gemini-2.0-flash) | – | – | 47.6% | 1.74e−2 | 9.56e−1 | 1.55e0 |
| MOL-LLM | 449 | 14306 | **100.0%** | 1.69e0 | 1.44e0 | 7.24e−1 |
| MOL-LLM (large dataset) | 15000 | 35915 | **100.0%** | 4.87e0 | 1.58e0 | 6.44e−1 |
| LLM (GPT-4.1) | – | – | **100.0%** | 7.79e−1 | 9.82e−1 | 1.00e0 |
| LLM (GPT-4o) | – | – | 85.0% | 9.24e−1 | 1.30e0 | 1.00e0 |
| LLM (Gemini-2.0-flash) | – | – | 33.3% | 9.02e−1 | `failed` | `failed` |
| OpInf-LLM (tool, GPT-4.1) | 449 | 30 | 99.2% | **1.29e−2** | 4.91e−1 | 4.63e−2 |
| OpInf-LLM (tool, GPT-4o) | 449 | 30 | 99.2% | **1.29e−2** | 4.91e−1 | 4.63e−2 |
| OpInf-LLM (tool, Gemini-2.0-flash) | 449 | 30 | 99.2% | **1.29e−2** | 4.91e−1 | 4.63e−2 |
| OpInf-LLM (codegen, GPT-4.1) | 449 | 30 | 99.2% | **1.29e−2** | 4.91e−1 | 3.40e−2 |
| OpInf-LLM (codegen, GPT-4o) | 449 | 30 | 99.2% | 1.87e−2 | 4.91e−1 | **3.26e−2** |
| OpInf-LLM (codegen, Gemini-2.0-flash) | 449 | 30 | 99.2% | **1.29e−2** | 4.91e−1 | 3.88e−1 |

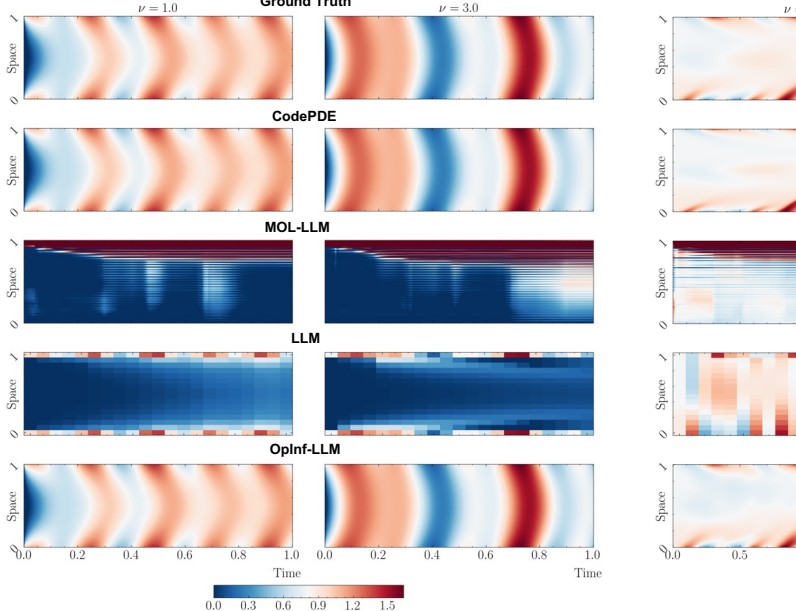

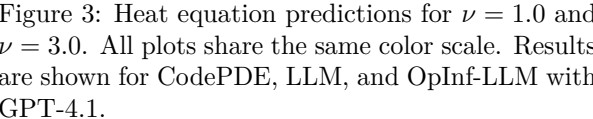

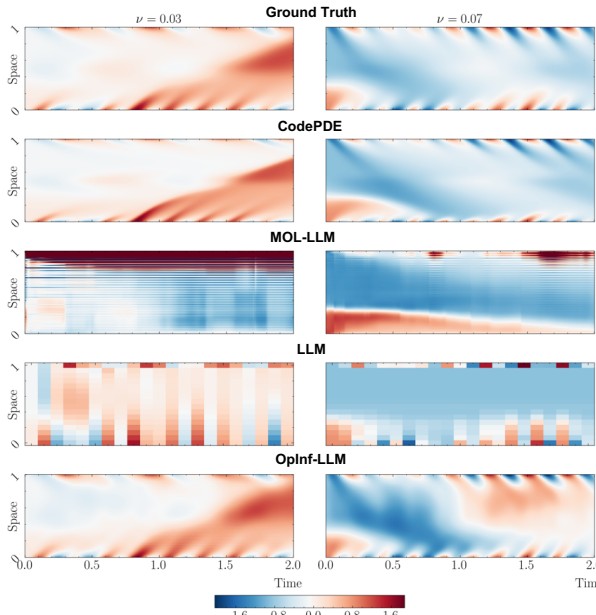

Figure 3: Heat equation predictions for $\nu = 1.0$ and $\nu = 3.0$. All plots share the same color scale. Results are shown for CodePDE, LLM, and OpInf-LLM with GPT-4.1.

Figure 4: Burgers' equation predictions for $\nu = 0.03$ and $\nu = 0.07$. All plots share the same color scale. Results are shown for CodePDE, LLM, and OpInf-LLM with GPT-4.1.

boundary conditions across different instances and previously unseen parameter configurations. The Burgers' equation test is particularly challenging due to its nonlinear advective dynamics and sensitivity to boundary conditions, and prediction errors are generally higher for all methods on this problem. It is also the only setting in which OpInf-LLM exhibits a failure case, where a specific boundary condition leads to an unstable reduced-order model.

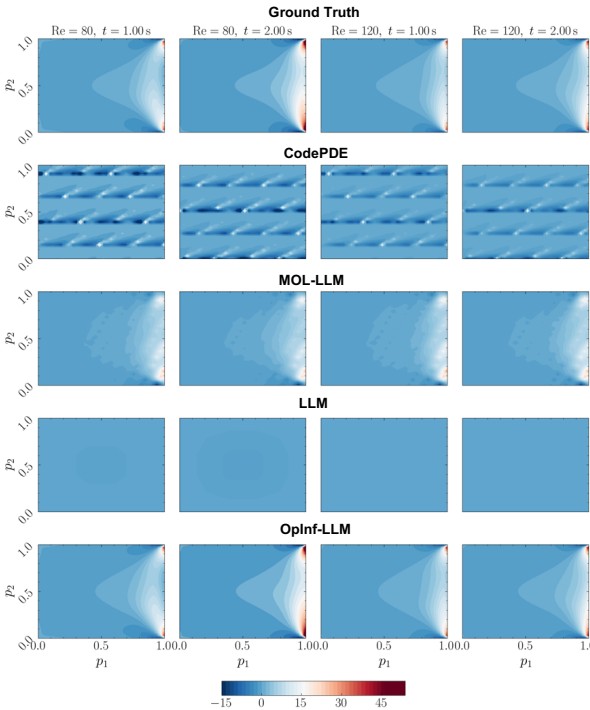

Figure 5: Lid-driven cavity predictions for Re = 80 and Re = 120 at $t = 1$ and $t = 2$. All plots share the same color scale. Results are shown for CodePDE, LLM and OpInf-LLM with GPT-4.1.

Table 2: Prediction error of OpInf-LLM with extrapolation in time.

| LLM model | Equation | Error $[0, T]$ | Error $[T, 2T]$ |
|---|---|---|---|
| GPT-4o | Heat | 1.29e−2 | 3.60e−2 |
| | Burgers | 4.91e−1 | 6.36e−1 |
| | Cavity | 4.63e−2 | 4.86e−2 |
| Gemini-2.0 flash | Heat | 1.29e−2 | 3.60e−2 |
| | Burgers | 4.91e−1 | 6.36e−1 |
| | Cavity | 4.63e−2 | 4.86e−2 |

## 4.4 Ablations

**Extrapolation in time.** We examine the capability of OpInf-LLM to extrapolatively predict solutions beyond the time horizons observed during training. Specifically, we evaluate its performance on the interval $[T, 2T]$ for the three equations considered, following the settings in Section 4.1, and report the relative $L_2$ error in Table 2. Although the framework has never seen any data on $[T, 2T]$, the prediction error remains consistently comparable to that on $[0, T]$, demonstrating strong temporal generalization of the proposed method. Additional visualizations on extended horizons are provided in Fig. 10–13 in the Appendix.

**Physically consistent operator feature identification.** We provide an additional ablation to evaluate whether OpInf-LLM can identify operator features that are consistent with the physical structure of the governing equation. In standard parametric OpInf, simple regression structures (e.g., linear dependence on parameters) are often assumed *a priori* (McQuarrie et al., 2023). However, the appropriate parameterization is problem-dependent and should reflect the underlying physics.

As a representative example, in the lid-driven cavity flow problem, the reduced operator depends linearly on 1/Re rather than Re. We evaluate whether OpInf-LLM can identify such a parameterization without

Table 3: Effect of feature parameterization on generalization for the lid-driven cavity problem.

| Method | Test Re | Distribution | Feature | Error |
|--------|---------|--------------|---------|-------|
| OpInf | 110 | In-distribution | Re | 3.57e−2 |
| OpInf-LLM | 110 | In-distribution | 1/Re | 2.57e−2 |
| OpInf | 200 | OOD | Re | instability |
| OpInf-LLM | 200 | OOD | 1/Re | 7.47e−2 |

Table 4: Natural language parsing performance comparison.

| Method | Exact Match | Field Accuracy |
|--------|-------------|----------------|
| Static Parser | 52% (26/50) | 75.7% (109/144) |
| OpInf-LLM | **100% (50/50)** | **100% (144/144)** |

equation-specific hints. To this end, we use a generic prompt: *"Define a helper parameter feature mapping for regression and choose a feature that yields smoother, better-conditioned operator trends."*

The results are summarized in Table 3. We observe that OpInf-LLM selects 1/Re as the feature, leading to improved accuracy compared to using Re directly. We use the same training setting as in Section 4.1, with parameter values up to Re = 150. For an out-of-distribution (OOD) test at Re = 200, the model achieves a relative $L_2$ error of 7.47e−2, which remains comparable to the in-distribution error. In contrast, using Re directly leads to instability in the OOD setting. Even within the training distribution, the improved parameterization yields lower prediction error.

Additional examples of physically consistent operator feature identification by OpInf-LLM are provided in Appendix C.4.

**Interpreting natural language instructions.** We then conduct ablation experiments to demonstrate that OpInf-LLM can robustly interpret diverse and unstructured natural language descriptions of PDE problems, compared to traditional structured parsers. Specifically, we evaluate OpInf-LLM against a static parser on 50 natural language scenarios describing PDEs and their parameter settings. The quantitative results are summarized in Table 4. The static parser frequently fails when handling implicit descriptions, informal phrasing and underspecified requests. In contrast, OpInf-LLM consistently maps these inputs to the correct PDE types and parameter configurations. For example, it correctly interprets *"temperature moving through a metal rod in a cooling line with diffusion rates 0.1 and 0.5"* as a heat equation with the corresponding diffusion coefficients. Experiment details and example failure cases for static solvers can be found in Appendix C.1.

## 5  Discussion

In this section, we discuss limitations and potential extensions of the proposed OpInf-LLM method. For a given PDE family, the same reduced basis can be reused across different parameter values and configurations, but new bases are usually constructed when introducing new governing equations, although such cost is incurred upfront offline rather than introducing additional computation at inference time. Another limitation is the dependence on the reduced-basis quality. For transport- or convection-dominated problems, sharp gradients, or far out-of-distribution boundary conditions, a low-dimensional linear basis may be insufficient, leading to degraded accuracy or instability (Dahmen et al., 2014; Mirhoseini & Zahr, 2023). A natural extension is to replace the linear POD representation with quadratic- or nonlinear-manifold OpInf formulations, which could improve the expressiveness of the reduced coordinates while preserving the language-guided OpInf-LLM workflow (Geelen et al., 2023; 2024). Notably, the OpInf framework does not require explicit knowledge of the full PDE form and only assumes the type of nonlinearity in the reduced dynamics, such as linear or quadratic structure, without needing the specific PDE instance or parameter values, which makes it well suited for settings in which the governing equations are partially unknown or difficult to identify. In addition, although we employ standard OpInf in our framework (Peherstorfer & Willcox, 2016), extensions of the OpInf

framework that improve robustness and accuracy (Sawant et al., 2023; Geng et al., 2024) could be used to further enhance the performance of OpInf-LLM.

## 6 Conclusions

In this work, we provide a new perspective on reduced-order modeling for LLM-based PDE solving, and propose OpInf-LLM, an LLM parametric PDE solving framework via operator inference. The proposed framework leverages shared reduced-order bases across each PDE family to support accurate, generalizable solutions under varying boundary conditions from limited data. By reformulating parametric PDE solving as polynomial regression in a reduced-order operator space followed by ODE integration, OpInf-LLM is able to solve unseen PDE instances from natural language instructions while achieving improved trade-offs between accuracy and execution success rate. Ablations further show robustness over extended test horizons and different OpInf configurations. Future directions include integrating more advanced operator inference techniques and developing richer language-solver interfaces for scalable scientific computing.

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

## A    Experiment Details

Below we provide additional details on input signal generation, numerical discretization, dataset construction, and simulation settings for the PDE systems described in Section 4.1.

### A.1    Heat Equation

The initial condition for the heat equation is fixed across all trajectories and given by

$$g(x) = \exp\big(\alpha(x-1)\big) + \exp(-\alpha x) - \exp(-\alpha), \qquad \alpha = 100. \tag{20}$$

The Dirichlet boundary input $u(t)$ applied at both ends of the domain is constructed as a three-component multi-sine signal

$$u(t) = b + \sum_{k=1}^{3} A_k \sin(2\pi f_k t + \phi_k), \tag{21}$$

where, for each trajectory, the amplitudes $A_k \sim \mathcal{U}(0.1, 0.5)$, frequencies $f_k \sim \mathcal{U}(0.2, 5.0)$, phases $\phi_k \sim \mathcal{U}(0, 2\pi)$, and bias $b \sim \mathcal{U}(0.8, 1.2)$ are sampled independently. Independent realizations of the boundary input are used for training and test datasets.

The spatial domain $x \in [0, 1]$ is discretized using 1023 uniformly spaced interior points, and the second-order central finite difference scheme is used to generate full-order PDE solution. Training trajectories are sampled at 1001 uniformly spaced time instances over the horizon $T = 1.0$, with BDF integrator. For evaluation, trajectories are generated over the extended horizon $T_{\text{test}} = 2.0$ using 2001 time steps.

### A.2    Burgers' Equation

The Burgers' equation includes an internal source term $s(x, t)$ with a fixed spatial profile and a time-dependent amplitude. The spatial component of the source is defined as

$$s_0(x) = \cosh\left(\frac{x - 0.5}{0.05}\right)^{-1}, \tag{22}$$

and the full source term is given by

$$s(x, t) = s_0(x)\, \sigma(t), \tag{23}$$

where the temporal modulation $\sigma(t)$ is generated as a single-component sinusoidal signal

$$\sigma(t) = b_s + A_s \sin(2\pi f_s t + \phi_s). \tag{24}$$

The parameters are sampled independently for each trajectory according to $A_s \sim \mathcal{U}(0.1, 3.0)$, $f_s \sim \mathcal{U}(0.2, 1.0)$, $\phi_s \sim \mathcal{U}(0, 2\pi)$, and $b_s \sim \mathcal{U}(-1.0, 1.0)$.

The system is subject to time-dependent Dirichlet boundary conditions

$$y(0, t) = u_1(t), \qquad y(1, t) = u_2(t), \tag{25}$$

where the boundary inputs $u_1(t)$ and $u_2(t)$ are generated independently as three-component multi-sine signals of the form

$$u_i(t) = b_i + \sum_{k=1}^{3} A_{i,k} \sin(2\pi f_{i,k} t + \phi_{i,k}), \qquad i \in \{1, 2\}. \tag{26}$$

For each trajectory, the amplitudes $A_{i,k} \sim \mathcal{U}(0.1, 1.0)$, frequencies $f_{i,k} \sim \mathcal{U}(0.2, 5.0)$, phases $\phi_{i,k} \sim \mathcal{U}(0, 2\pi)$, and biases $b_i \sim \mathcal{U}(-0.5, 0.5)$ are sampled independently.

The initial condition is constructed to be smooth and consistent with the boundary values at $t = 0$:

$$y(x, 0) = u_2(0) + \frac{1}{2}\big(u_1(0) - u_2(0)\big) \left[1 - \tanh\left(\frac{x - 0.3}{0.1}\right)\right]. \tag{27}$$

The spatial discretization uses Chebyshev collocation with $N = 64$ interior points. For evaluation and visualization, the solution is interpolated onto a uniform grid with 1001 points. Training trajectories span the time horizon $T = 2.0$ with 1001 time steps, while test trajectories span $T_{\text{test}} = 4.0$ with 2001 time steps. All source and boundary signals are sampled independently across trajectories. RK45 integrator is used for full-order solution generation.

### A.3  2D Lid-Driven Cavity Flow

The no-slip boundary conditions enforce zero velocity on all walls except the top lid. The horizontal velocity on the top boundary is prescribed as

$$v_1(x, t) = h(p_1) \, u(t), \qquad v_2(x, t) = 0, \tag{28}$$

where the fixed spatial profile is

$$h(p_1) = 1 + 0.3 \sin(2\pi p_1) + 0.2 \, p_1. \tag{29}$$

The temporal modulation $u(t)$ is generated as a single-sine signal

$$u(t) = b + A \sin(2\pi f t + \phi), \tag{30}$$

where $b \sim \mathcal{U}(0.7, 1.3)$, $A \sim \mathcal{U}(0.1, 0.4)$, $f \sim \mathcal{U}(0.5, 2.0)$, and $\phi \sim \mathcal{U}(0, 2\pi)$ are sampled independently for each trajectory.

The cavity domain is defined on $x = (p_1, p_2) \in [0, 1]^2$ and discretized using a uniform grid with 32 interior points in each spatial direction, resulting in a $(34 \times 34)$ grid including boundary points. The full-order solver uses a fixed time step $\Delta t = 10^{-3}$, with second-order finite difference scheme in space and forward Euler temporal integration. Training data are collected over the horizon $T = 2.0$, with solution snapshots recorded every 0.02 time units, yielding 101 snapshots per trajectory. For evaluation, trajectories are simulated over $T_{\text{test}} = 4.0$ using the same solver and snapshot spacing.

### A.4  Dataset Construction

For all PDE systems, training and test datasets are generated using independent random seeds for boundary and forcing inputs. Each dataset consists of multiple trajectories corresponding to different realizations of input signals and, where applicable, different physical parameter values.

All reported results are obtained using identical datasets across methods to ensure fair comparison. Numerical solvers, discretization schemes, and time-stepping parameters are fixed across training and testing, and no test-time information is used during model training.

## B  Implementation Details

### B.1  CodePDE

We evaluate CodePDE (Li et al., 2025) using 5 rounds of solver generation for each PDE, with 5 debug trials for each round. The test performance is measured using the relative $L_2$ error over the full time horizon. The best solver for each equation is reported for prediction errors. Success rate is calculated based on trial-level failures. The prompts and solver templates for all equations are shown below.

**Prompts:**

```
heat_description = '''
The PDE is the 1D heat equation with time-varying Dirichlet boundary conditions,
    given by

\\[
\\begin{{cases}}
```

```
\\partial_t u(t, x) = \\nu \\partial_{{xx}} u(t, x), & x \\in (0,1), \; t \\in
    (0,T] \\\\
u(0, x) = u_0(x), & x \\in (0,1) \\\\
u(t, 0) = u_{{bc}}(t), & t \\in (0,T] \\\\
u(t, 1) = u_{{bc}}(t), & t \\in (0,T]
\\end{{cases}}
\\]

where $\\nu$ is the thermal diffusivity coefficient. **Note that the boundary
    conditions are symmetric** (same function at both boundaries) and time-varying.

**Initial Condition (Fixed)**: The initial condition is a fixed exponential
    profile given by
\\[
u_0(x) = e^{{100(x-1)}} + e^{{-100x}} - e^{{-100}}
\\]
This creates steep gradients near the boundaries with approximate zero values at
    both ends.

**Boundary Conditions (Time-varying)**: The boundary value $u_{{bc}}(t)$ is
    provided as an input array of shape [batch_size, T+1] containing the boundary
    values at each time step. The same values apply to both $x=0$ and $x=1$
    (symmetric boundaries).

Given the discretization of the boundary condition values $u_{{bc}}(t)$ of shape
    [batch_size, T+1], you need to implement a solver that:
1. Computes the fixed initial condition for the given spatial grid
2. Solves the heat equation forward in time while enforcing the time-varying
    boundary conditions
3. Returns the solution of shape [batch_size, T+1, N] where N=1023 (interior
    spatial points)

In particular, your code should be tailored to the case where
    $\\nu={opinf_heat_nu}$, i.e., optimizing it particularly for this use case.

**Important implementation notes**:
- Use **1023 interior spatial points** (not including boundaries at x=0 and x=1)
- The domain is $x \\in [0,1]$ with $dx \\approx 9.766 \\times 10^{{-4}}$
- Time horizon is $T=1.0$ with 1001 time points ($dt = 10^{{-3}}$)
- The boundary conditions must be enforced at every time step
- Consider using implicit methods (e.g., BDF, Crank-Nicolson) for stability
- Use finite difference for spatial discretization (2nd order central differences
    recommended)
'''

burgers_description = '''
The PDE is the 1D Burgers equation with asymmetric time-varying Dirichlet
    boundary conditions and a localized source term, given by

\\[
\\begin{{cases}}
\\partial_t u(t, x) + u(t, x) \\partial_x u(t, x) = \\nu \\partial_{{xx}} u(t, x)
    + s(x) \\cdot w_3(t), & x \\in (0,1), \; t \\in (0,T] \\\\
u(0, x) = u_0(x), & x \\in (0,1) \\\\
u(t, 0) = w_1(t), & t \\in (0,T] \\\\
u(t, 1) = w_2(t), & t \\in (0,T]
\\end{{cases}}
\\]
```

```
where $\\nu$ is the viscosity coefficient, and we have **three time-varying input
    functions** provided as arrays:
- $w_1(t)$: Left boundary condition (shape [batch_size, T+1])
- $w_2(t)$: Right boundary condition (shape [batch_size, T+1])
- $w_3(t)$: Source term temporal modulation (shape [batch_size, T+1])

**Initial Condition (Variable)**: The initial condition depends on the boundary
    values at $t=0$ and must be computed:
\\[
u_0(x) = w_2(0) + 0.5(w_1(0) - w_2(0))(1 - \\tanh((x - 0.3)/0.1))
\\]
This creates a smooth transition from the left BC to the right BC, centered at
    $x=0.3$ with transition width $0.1$.

**Source Term (Localized)**: The spatial source profile is a narrow bell-shaped
    function centered at $x=0.5$:
\\[
s(x) = \\text{{sech}}^2\\left(\\frac{{x - 0.5}}{{0.05}}\\right) =
    \\frac{{1}}{{\\cosh^2\\left(\\frac{{x - 0.5}}{{0.05}}\\right)}}
\\]
The full source term is $s(x) \\cdot w_3(t)$, where $w_3(t)$ modulates the source
    strength in time (element-wise multiplication for each spatial point).

**Boundary Conditions (Asymmetric, Time-varying)**: The left and right boundaries
    have **independent** time-varying values $w_1(t) \\neq w_2(t)$.

Given the discretization of the boundary functions $w_1(t)$, $w_2(t)$, and source
    modulation $w_3(t)$ (each of shape [batch_size, T+1]), you need to implement a
    solver that:
1. Computes the initial condition from $w_1(0)$ and $w_2(0)$
2. Evaluates the spatial source profile $s(x)$ on the grid
3. Solves the Burgers equation forward in time with boundary conditions and
    source term
4. Returns the solution of shape [batch_size, T+1, N] where N is the number of
    spatial grid points

In particular, your code should be tailored to the case where
    $\\nu={opinf_burgers_nu}$, i.e., optimizing it particularly for this use case.

**Important implementation notes**:
- Spatial discretization: 1001 points uniformly spaced in [0,1] (including
    boundaries) or use spectral methods
- Time horizon is $T=2.0$ with 1001 time points ($dt = 2 \\times 10^{{-3}}$)
- The source term is localized at the center ($x=0.5$) with very narrow width
    (0.05)
- Boundary conditions are **asymmetric**: $w_1(t) \\neq w_2(t)$
- The nonlinear convection term $u \\partial_x u$ can cause shocks/steep
    gradients at low viscosity
- Consider using explicit methods (e.g., RK45) or spectral methods for accuracy
- Pay special attention to handling the localized source term accurately
'''

cavity_description = '''
The PDE is the 2D lid-driven cavity flow in vorticity-streamfunction form:

\\[
\\omega_t + u \\omega_x + v \\omega_y = \\frac{{1}}{{Re}}(\\omega_{{xx}} +
    \\omega_{{yy}}), \\quad (x, y) \\in (0,1)^2
```

```
\\]
\\[
\\Delta \\psi = -\\omega, \\quad u = \\psi_y, \\quad v = -\\psi_x
\\]

Boundary conditions are no-slip on all walls. The top lid has a time-varying
    velocity:
\\[
u_{{lid}}(x, t) = a(x) f(t), \\quad v=0
\\]
where the fixed spatial profile is:
\\[
a(x) = 1 + 0.3\\sin(2\\pi x) + 0.2x
\\]
and the input is a scalar function f(t).

Given the discretization of f(t) as an input array of shape [batch_size, T+1],
    you need to implement
a solver that returns the vorticity field \\omega(t, x, y) for all times. The
    solution should be of shape
[batch_size, T+1, N, N] where N=34 (32 interior points plus boundaries).

In particular, your code should be tailored to the case where
    Re={opinf_cavity_re}.

Important implementation notes:
- Initial condition: \\omega(x, y, 0) = 0 and \\psi(x, y, 0) = 0
- Spatial grid: 34x34 uniform grid on [0, 1]^2
- Time step in the full-order model is dt = 0.001 with snapshots every 0.02
- Use finite differences for spatial derivatives and a Poisson solver for \\psi
- Consider small internal time steps for stability
'''
```

**Solver templates:**

```python
import numpy as np

def solver(u_bc, t_coordinate, nu):
    """Solves the 1D heat equation with time-varying Dirichlet boundary conditions.

    Args:
        u_bc (np.ndarray): Boundary condition values at both x=0 and x=1
            Shape: [batch_size, T+1] where T+1 is the number of time points.
            The same boundary value applies to both boundaries (symmetric).
        t_coordinate (np.ndarray): Time coordinates of shape [T+1].
            It begins with t_0=0 and follows the time steps t_1, ..., t_T.
        nu (float): Thermal diffusivity coefficient.

    Returns:
        solutions (np.ndarray): Shape [batch_size, T+1, N] where N=1023 (interior points).
            solutions[:, 0, :] contains the initial conditions (fixed exponential profile),
            solutions[:, i, :] contains the solutions at time t_coordinate[i].
    """
    # TODO: Implement the solver for the heat equation with time-varying BCs
    #
    # Key implementation steps:
    # 1. Create spatial grid: N=1023 interior points in [0, 1]
    #     x = linspace(0, 1, 1025)[1:-1]  # Interior points only
    #     dx = 1/1024
    #
    # 2. Compute fixed initial condition:
    #     u0 = exp(100*(x-1)) + exp(-100*x) - exp(-100)
```

```python
29        #
30        # 3. Set up finite difference matrix for d^2/dx^2 (2nd order central)
31        #
32        # 4. Time integration loop:
33        #    - Enforce boundary conditions at each time step
34        #    - Solve: du/dt = nu * d^2 u/dx^2
35        #    - Consider using implicit methods (BDF, Crank-Nicolson) for stability
36        #
37        # Hints:
38        # - Use scipy.integrate.solve_ivp with method='BDF' or implement implicit scheme
39        # - Consider using PyTorch or JAX for GPU acceleration
40        # - Remember that u_bc has shape [batch_size, T+1] with same values for both boundaries
41
42        return solutions

46  def solver(w1_bc, w2_bc, source, t_coordinate, nu):
47        """Solves the 1D Burgers equation with asymmetric time-varying Dirichlet BCs and source
          term.
48
49        Args:
50            w1_bc (np.ndarray): Left boundary condition values at x=0
51                Shape: [batch_size, T+1] where T+1 is the number of time points.
52            w2_bc (np.ndarray): Right boundary condition values at x=1
53                Shape: [batch_size, T+1]. Note: w1 \neq w2 (asymmetric boundaries).
54            source (np.ndarray): Source term temporal modulation w3(t)
55                Shape: [batch_size, T+1]. This modulates the spatial source profile s(x).
56            t_coordinate (np.ndarray): Time coordinates of shape [T+1].
57                It begins with t_0=0 and follows the time steps t_1, ..., t_T.
58            nu (float): Viscosity coefficient.
59
60        Returns:
61            solutions (np.ndarray): Shape [batch_size, T+1, N] where N is spatial grid points.
62                solutions[:, 0, :] contains the initial conditions (computed from w1(0), w2(0)),
63                solutions[:, i, :] contains the solutions at time t_coordinate[i].
64        """
65        # TODO: Implement the solver for Burgers equation with BCs and source term
66        #
67        # Key implementation steps:
68        # 1. Create spatial grid: N=1001 points (including boundaries) or use spectral
69        #     x = linspace(0, 1, 1001) for uniform grid
70        #     dx = 1/1000
71        #
72        # 2. Compute spatial source profile s(x):
73        #     s_x = 1 / cosh^2((x - 0.5) / 0.05)
74        #     This is a narrow bell-shaped function centered at x=0.5
75        #
76        # 3. Compute initial condition for each batch item:
77        #     u0 = w2_bc[:, 0] + 0.5 * (w1_bc[:, 0] - w2_bc[:, 0]) * (1 - tanh((x - 0.3) / 0.1))
78        #     Note: u0 depends on boundary values at t=0
79        #
80        # 4. Time integration loop:
81        #    - Enforce boundary conditions at each time step:
82        #       u[:, 0] = w1_bc[:, t_idx]   (left)
83        #       u[:, -1] = w2_bc[:, t_idx]   (right)
84        #    - Solve: du/dt + u*du/dx = nu*d^2 u/dx^2 + s(x)*source[:, t_idx]
85        #    - Consider using RK45 or spectral methods for accuracy
86        #
87        # Hints:
88        # - The source term is localized (narrow width 0.05) - handle carefully
89        # - Boundaries are asymmetric: w1(t) \neq w2(t)
90        # - Consider using scipy.integrate.solve_ivp with method='RK45'
91        # - For spectral methods, use Chebyshev collocation
92        # - Nonlinear term u*du/dx can cause shocks at low viscosity
93
94        return solutions
95
```

```
96
97  def solver(lid_bc, t_coordinate, re):
98      """Solves 2D lid-driven cavity flow in vorticity-streamfunction form.
99
100     Args:
101         lid_bc (np.ndarray): Scalar lid velocity modulation f(t).
102             Shape: [batch_size, T+1]. The top-wall velocity is a(x) * f(t),
103             where a(x) is a fixed spatial profile defined in the spec.
104         t_coordinate (np.ndarray): Time coordinates of shape [T+1].
105             It begins with t_0=0 and follows the time steps t_1, ..., t_T.
106         re (float): Reynolds number.
107
108     Returns:
109         solutions (np.ndarray): Shape [batch_size, T+1, N, N] where N=34 (includes
        boundaries).
110             solutions[:, 0, :, :] contains the initial vorticity (zeros),
111             solutions[:, i, :, :] contains the vorticity at time t_coordinate[i].
112     """
113     # TODO: Implement the cavity flow solver in vorticity-streamfunction form.
114     #
115     # Key implementation steps:
116     # 1. Create spatial grid: N=34 (32 interior + 2 boundaries) in [0, 1] x [0, 1]
117     # 2. Initialize vorticity omega = 0 and streamfunction psi = 0
118     # 3. Enforce no-slip boundaries; top lid velocity: u_lid(x, t) = a(x) * f(t)
119     #    with a(x) = 1 + 0.3*sin(2*pi*x) + 0.2*x
120     # 4. Time integration loop:
121     #    - Solve Poisson: Laplacian(psi) = -omega
122     #    - Compute u = psi_y, v = -psi_x
123     #    - Advance omega with advection-diffusion: omega_t + u*omega_x + v*omega_y = (1/re)
        * Laplacian(omega)
124     # 5. Return vorticity snapshots over time
125     #
126     # Hints:
127     # - Use finite differences for spatial derivatives (2nd order central)
128     # - Consider an explicit time step (dt <= 1e-3) for stability
129     # - Use a fast Poisson solver (e.g., FFT or sparse solve) for psi
130
131     return solutions
```

## B.2  MOL-LLM

We implement MOL-LLM (Negrini et al., 2025) based on the original implementation[3], with extensions to support time-varying boundary conditions and 2D PDEs. The model adopts a GPT-2–style transformer backbone that processes a concatenation of natural-language PDE descriptions and a parallel numeric token stream encoding the initial state, PDE coefficients, and boundary condition (BC) parameters. In addition to the standard coefficient embedding (e.g., viscosity $\nu$ or Reynolds number $Re$), we add the 34-dimensional BC parameter vector that encodes multi-sine boundary or forcing signals in equation 21, equation 26, equation 30 (zero-filled when inactive). This vector is embedded by a dedicated MLP into the same token space and treated as an additional metadata token so the transformer can explicitly attend to BC information. Training is performed over a fixed horizon $[0, T]$ with $t_{\text{len}} = 128$ time points, and solution values are produced by a time-query head conditioned on the initial state and metadata tokens. For the 2D lid-driven cavity equation, the vorticity field is flattened to a one-dimensional vector and downsampled to 128 spatial points to match the model's maximum numeric dimension. Predictions are interpolated back to the original $34 \times 34$ grid using inverse-distance weighting for full-order model comparison and visualization. The combined transformer input consists of GPT-2 tokenized text (augmented with number and padding tokens) concatenated with numeric tokens corresponding to the initial state, coefficient vector, and the BC token, enabling unified attention over symbolic PDE descriptions and structured numeric conditioning. The text descriptions for each equation used for training is below.

**Text Descriptions:**

---

[3]https://github.com/enegrini/MOL-LLM

```
{"type": 60, "text": [["The parametric heat equation is ut = nu * uxx with
    time-varying boundary conditions and varying viscosity. We have initial
    condition ="], ["and viscosity coefficient nu ="], ["This equation models
    diffusion with time-varying boundary forcing."]]}
{"type": 60, "text": [["The parametric heat equation is ut = nu * uxx with
    time-varying boundary conditions and varying viscosity. We have initial
    condition ="], ["and viscosity coefficient nu ="], ["The viscosity parameter
    controls the rate of heat diffusion in the domain."]]}
{"type": 60, "text": [["The parametric heat equation is ut = nu * uxx with
    time-varying boundary conditions and varying viscosity. We have initial
    condition ="], ["and viscosity coefficient nu ="], ["This formulation captures
    1D heat transport with parametric viscosity."]]}
{"type": 60, "text": [["The parametric heat equation is ut = nu * uxx with
    time-varying boundary conditions and varying viscosity. We have initial
    condition ="], ["and viscosity coefficient nu ="], ["The solution evolves
    under diffusive dynamics driven by boundary inputs."]]}
{"type": 60, "text": [["The parametric heat equation is ut = nu * uxx with
    time-varying boundary conditions and varying viscosity. We have initial
    condition ="], ["and viscosity coefficient nu ="], ["This case studies
    diffusion with parametric viscosity and time-varying BCs."]]}

{"type": 61, "text": [["The parametric Burgers equation is ut + u*ux = nu*uxx +
    s(x)*w3(t) with time-varying boundary conditions. We have initial condition
    ="], ["and viscosity coefficient nu ="], ["This equation models nonlinear
    advection-diffusion with a time-dependent forcing."]]}
{"type": 61, "text": [["The parametric Burgers equation is ut + u*ux = nu*uxx +
    s(x)*w3(t) with time-varying boundary conditions. We have initial condition
    ="], ["and viscosity coefficient nu ="], ["The viscosity controls shock
    smoothing in the nonlinear transport dynamics."]]}
{"type": 61, "text": [["The parametric Burgers equation is ut + u*ux = nu*uxx +
    s(x)*w3(t) with time-varying boundary conditions. We have initial condition
    ="], ["and viscosity coefficient nu ="], ["This setup couples boundary driving
    and interior forcing in 1D flow."]]}
{"type": 61, "text": [["The parametric Burgers equation is ut + u*ux = nu*uxx +
    s(x)*w3(t) with time-varying boundary conditions. We have initial condition
    ="], ["and viscosity coefficient nu ="], ["Nonlinear advection competes with
    diffusion and time-varying inputs."]]}
{"type": 61, "text": [["The parametric Burgers equation is ut + u*ux = nu*uxx +
    s(x)*w3(t) with time-varying boundary conditions. We have initial condition
    ="], ["and viscosity coefficient nu ="], ["This equation demonstrates driven
    viscous Burgers dynamics."]]}

{"type": 62, "text": [["The 2D lid-driven cavity flow in vorticity-streamfunction
    formulation omega_t + u*omega_x + v*omega_y = (1/Re)*(omega_xx + omega_yy)
    models incompressible viscous flow in a square cavity with time-varying
    spatially-varying lid velocity. We have initial condition ="], ["and Reynolds
    number Re ="], ["The Reynolds number controls the relative importance of
    inertial versus viscous effects in the flow."]]}
{"type": 62, "text": [["The 2D lid-driven cavity flow in vorticity-streamfunction
    formulation omega_t + u*omega_x + v*omega_y = (1/Re)*(omega_xx + omega_yy)
    models incompressible viscous flow in a square cavity with time-varying
    spatially-varying lid velocity. We have initial condition ="], ["and Reynolds
    number Re ="], ["This equation captures cavity circulation driven by a moving
    lid."]]}
{"type": 62, "text": [["The 2D lid-driven cavity flow in vorticity-streamfunction
    formulation omega_t + u*omega_x + v*omega_y = (1/Re)*(omega_xx + omega_yy)
    models incompressible viscous flow in a square cavity with time-varying
    spatially-varying lid velocity. We have initial condition ="], ["and Reynolds
```

```
     number Re ="], ["Higher Reynolds number increases inertial effects in the
     cavity flow."]]}
{"type": 62, "text": [["The 2D lid-driven cavity flow in vorticity-streamfunction
     formulation omega_t + u*omega_x + v*omega_y = (1/Re)*(omega_xx + omega_yy)
     models incompressible viscous flow in a square cavity with time-varying
     spatially-varying lid velocity. We have initial condition ="], ["and Reynolds
     number Re ="], ["This setup models 2D vortical dynamics with time-varying lid
     motion."]]}
{"type": 62, "text": [["The 2D lid-driven cavity flow in vorticity-streamfunction
     formulation omega_t + u*omega_x + v*omega_y = (1/Re)*(omega_xx + omega_yy)
     models incompressible viscous flow in a square cavity with time-varying
     spatially-varying lid velocity. We have initial condition ="], ["and Reynolds
     number Re ="], ["The vorticity-streamfunction form enforces incompressibility
     in the cavity."]]}
```

### B.3  LLMs

We directly prompt large language models with the governing PDE, full initial and boundary conditions, external inputs, and a coarse output grid for tractability (heat/Burgers: 16 spatial points $\times$ 41 time steps; cavity: $6 \times 6$ spatial points $\times$ 21 time steps), and ask it to return the full spatiotemporal state array in JSON format (3-decimal precision) for a single parameter setting and trajectory. The LLM is treated as a black-box simulator, and its predicted field is compared against downsampled ground truth data using the relative $L_2$ error. For each parameter instance and equation, the LLM is queried 10 times, and the averaged error over all successful cases is reported. Example prompts are provided below.

**Prompts:**

```
Heat:

You are given a PDE, full IC/BC specs, parameter values, and input signals.
     Return ONLY a valid JSON object with numeric arrays at 3-decimal precision.
     No extra text, no code fences.
Task: Heat equation u_t = nu * u_xx on x in [0,1].\nBoundary conditions: u(0,t)=
     u(1,t)=u_bc(t).\nParameter: nu = 0.5.\nOutput grid: 16 spatial points, 41
     time steps.\nReturn JSON with fields: equation, param, grid_shape, t_steps, x
     , t, u0, u.\nAll numeric values must be rounded to 3 decimals.\nx = [0.001,
     0.067,0.134,0.200,0.268,0.334,0.400,0.467,0.533,0.600,0.666,0.732,0.800,0.866
     ,0.933,0.999]\nt = [0.000,0.050,0.100,0.150,0.200,0.250,0.300,0.350,0.400,
     0.450,0.500,0.550,0.600,0.650,0.700,0.750,0.800,0.850,0.900,0.950,1.000,1.050
     ,1.100,1.150,1.200,1.250,1.300,1.350,1.400,1.450,1.500,1.550,1.600,1.650,
     1.700,1.750,1.800,1.850,1.900,1.950,2.000]\nu0(x) = [0.581,0.001,0.000,0.000,
     0.000,0.000,0.000,0.000,0.000,0.000,0.000,0.000,0.000,0.000,0.001,0.581]\
     nu_bc(t) = [0.581,0.762,0.783,0.686,0.648,0.818,1.184,1.564,1.759,1.705,1.527
     ,1.418,1.473,1.605,1.625,1.409,1.019,0.655,0.498,0.565,0.705,0.736,0.609,
     0.452,0.456,0.709,1.109,1.442,1.558,1.483,1.391,1.440,1.632,1.804,1.773,1.491
     ,1.099,0.810,0.737,0.812,0.848]\nJSON schema:\n{"equation":"heat","param":{"
     nu":...},"grid_shape":[16],"t_steps":41,"x":[...],"t":[...],"u0":[...],"u
     ":[[...]]}\n

Burgers:

You are given a PDE, full IC/BC specs, parameter values, and input signals.
     Return ONLY a valid JSON object with numeric arrays at 3-decimal precision.
     No extra text, no code fences.
Task: Burgers equation u_t + u*u_x = nu*u_xx + s(x)*w3(t), x in [0,1].\nBoundary
     : u(0,t)=w1(t), u(1,t)=w2(t).\nForcing shape: s(x)=cosh((x-0.5)/0.05)^(-1).\
     nParameter: nu = 0.03.\nOutput grid: 16 spatial points, 41 time steps.\
```

```
     nReturn JSON with fields: equation, param, grid_shape, t_steps, x, t, u0, w1,
      w2, w3, u.\nAll numeric values must be rounded to 3 decimals.\nx = [0.000,
     0.067,0.133,0.200,0.267,0.333,0.400,0.467,0.533,0.600,0.667,0.733,0.800,0.867
     ,0.933,1.000]\nt = [0.000,0.100,0.200,0.300,0.400,0.500,0.600,0.700,0.800,
     0.900,1.000,1.100,1.200,1.300,1.400,1.500,1.600,1.700,1.800,1.900,2.000,2.100
     ,2.200,2.300,2.400,2.500,2.600,2.700,2.800,2.900,3.000,3.100,3.200,3.300,
     3.400,3.500,3.600,3.700,3.800,3.900,4.000]\nu0(x) = [0.084,0.082,0.077,0.060,
     0.015,-0.050,-0.095,-0.112,-0.117,-0.119,-0.119,-0.119,-0.119,-0.119,-0.119,
     -0.119]\nw1(t) = [0.084,1.099,-1.020,-0.635,-0.467,-1.006,1.104,-0.197,1.551,
     0.562,0.421,0.978,-0.573,1.219,-0.322,0.766,0.327,-0.576,0.454,-1.390,0.491,
     -0.253,0.836,1.754,0.708,2.081,-0.317,0.278,-0.945,-1.321,0.040,-0.923,1.531,
     0.528,1.461,1.244,-0.077,1.020,-0.974,0.595,-0.267]\nw2(t) = [-0.119,-0.546,
     0.115,1.417,0.645,0.414,-0.283,-0.293,0.474,1.305,0.912,-0.492,-0.064,0.006,
     0.848,1.094,0.622,-0.517,-0.451,0.901,0.789,0.948,0.092,-0.363,-0.286,1.031,
     1.299,0.143,0.010,-0.400,0.336,0.927,1.299,-0.045,-0.603,0.244,0.519,1.156,
     0.654,0.035,-0.790]\nw3(t) = [-2.453,-1.108,0.170,1.133,1.618,1.573,1.077,
     0.313,-0.470,-1.018,-1.133,-0.719,0.192,1.447,2.806,3.992,4.757,4.940,4.499,
     3.528,2.232,0.885,-0.234,-0.903,-1.007,-0.566,0.271,1.259,2.112,2.570,2.457,
     1.724,0.465,-1.107,-2.703,-4.019,-4.806,-4.923,-4.371,-3.288,-1.923]\nJSON
     schema:\n{"equation":"burgers","param":{"nu":...},"grid_shape":[16],"t_steps
     ":41,"x":[...],"t":[...],"u0":[...],"w1":[...],"w2":[...],"w3":[...],"u
     ":[[...]]}\n

Cavity:

You are given a PDE, full IC/BC specs, parameter values, and input signals.
    Return ONLY a valid JSON object with numeric arrays at 3-decimal precision.
    No extra text, no code fences.
Task: 2D lid-driven cavity (vorticity/streamfunction).\nEquations: omega_t + u*
    omega_x + v*omega_y = (1/Re)*(omega_xx+omega_yy), Laplacian(psi) = -omega, u=
    psi_y, v=-psi_x.\nNo-slip walls. Lid velocity u_lid(x,t)=a(x)*f(t), v=0.\na(x
    )=1+0.3*sin(2*pi*x)+0.2*x (fixed).\nInitial condition: omega=0, psi=0.\
    nParameter: Re = 120.\nOutput grid: 6x6, 21 time steps.\nReturn JSON with
    fields: equation, param, grid_shape, t_steps, x, y, t, u_lid, omega, psi.\
    nAll numeric values must be rounded to 3 decimals.\nx = [0.000,0.212,0.394,
    0.606,0.788,1.000]\ny = [0.000,0.212,0.394,0.606,0.788,1.000]\nt = [0.000,
    0.200,0.400,0.600,0.800,1.000,1.200,1.400,1.600,1.800,2.000,2.180,2.380,2.580
    ,2.780,2.980,3.180,3.380,3.580,3.780,3.980]\nu_lid(t) = [0.658,0.728,0.873,
    0.606,0.826,0.793,0.618,0.890,0.693,0.689,0.891,0.639,0.754,0.856,0.602,0.847
    ,0.767,0.632,0.897,0.670,0.714]\nJSON schema:\n{"equation":"cavity","param
    ":{"Re":...},"grid_shape":[6,6],"t_steps":21,"x":[...],"y":[...],"t":[...],"
    u_lid":[...],"omega":[[[...]]],"psi":[[[...]]]}\n
```

## B.4 OpInf-LLM

We collect data and produce reduced-order operators following the settings in Section 4.1, and prompt an LLM to predict reduced-order operators at new parameter values by interpolating or regressing from a set of training parameters with known operators. Specifically, the LLM generates code or invokes a local interpolation or regression tool to compute the reduced-order operators at the queried parameter value, and optionally validates them via lightweight heuristics. The predicted operators are then used to simulate the reduced-order model (ROM). The key procedures are summarized in Alg. 1. Relative $L_2$ errors are reported over both the training time horizon and an extended time horizon to assess long-term generalization. System prompts and the provided tools used by OpInf-LLM are listed below.

---

**Algorithm 1** OpInf-LLM

---

**Given:** basis number $r$, regularization intensity $\lambda$

**Offline:**

1: Collect PDE solution data $\mathcal{D}$ as in equation 9 of different parameters $\xi$, from multiple trajectories with different ICBCs.

2: Identify basis $\Phi := [\phi_1(\boldsymbol{x}), \dots, \phi_r(\boldsymbol{x})]$ through POD.
Specifically, $\mathbf{Y} = \mathbf{U}\boldsymbol{\Sigma}\mathbf{V}^\top$ with

$$\mathbf{Y} = [y(\boldsymbol{x}, t_1, \xi_1), \dots, y(\boldsymbol{x}, t_K, \xi_1), y(\boldsymbol{x}, t_1, \xi_2), \dots, y(\boldsymbol{x}, t_K, \xi_2), \dots]$$

and $\mathbf{U} = [\phi_1, \cdots, \phi_r, \cdots]$.

3: Identify reduced-order operators $A$, $H$, $B$, $c$ for all parameters $\xi$ in the dataset via equation 8.

**Online:**

4: Parse natural language PDE solving instruction to obtain parameter $\xi$ and ICBC of interest.

5: Perform interpolation or regression to identify reduced operators $A$, $H$, $B$, $c$ for test parameter $\xi$.

6: Integrate ODE to solve reduced dynamics in equation 4.

7: Reconstruct full-order solution via equation 3.

---

**Prompts:**

```
Parsing:

schema = {
        "equations": ["heat", "burgers", "cavity"],
        "provider": "openai|gemini|deepseek|anthropic|qwen",
        "model_name": "string or null",
        "output_dir": "string or null",
        "save_raw": "true/false",
        "reuse_operators": "true/false",
        "heat_nus": "list of numbers or null",
        "burgers_nus": "list of numbers or null",
        "cavity_res": "list of numbers or null",
    }
    system = (
        "You are a strict parser. Return JSON only. Do not include code fences."
    )
    user = (
        "Parse the prompt into the following JSON keys. Use lowercase for
            equations.\n"
        f"Schema: {json.dumps(schema)}\n"
        f"Prompt: {prompt}"
    )
    messages = [
        {"role": "system", "content": system},
        {"role": "user", "content": user},
    ]

Solving:

**Task**: Predict OpInf operators for a {equation_type} equation at parameter \
    nu = {query_nu}

**Available Data** (already loaded in the system):
- Training parameters: \nu = {nu_train}
- Operators at each \nu: {operator_names}
```

```
**Available Tools** (USE THESE IN ORDER):
1. 'analyze_parameter_range(nu_train, nu_query)': Check if \nu={query_nu} is
    interpolation or extrapolation
{step_2}
3. 'validate_operators(operators, equation_type)': Check physical constraints

**Instructions**:
1. First, call analyze_parameter_range({nu_train}, {query_nu}) to understand the
    problem
{instruction_2}
3. Finally, call validate_operators with the returned operators and
    equation_type="{equation_type}"

Please solve this step-by-step using the tools.

if method == "regression":
    step_2 = f"2. 'simple_linear_regress(nu_query)': Get regressed operators (
        just pass nu_query={query_nu})"
    instruction_2 = f"2. Then, call simple_linear_regress({query_nu}) to get the
        operators"
else:
    step_2 = f"2. 'simple_interpolate(nu_query, method)': Get interpolated
        operators (EASIEST - just pass nu_query={query_nu})"
    instruction_2 = f"2. Then, call simple_interpolate({query_nu}, \"linear\")
        to get the operators"

initial_prompt = f"""You are an expert in reduced-order modeling and operator
    inference.
"""
```

**Provided tools:**

```
 1
 2  # ==============================================================================
 3  # Tool Functions
 4  # ==============================================================================
 5
 6  def interpolate_operators(
 7      nu_train: List[float],
 8      operators_train: Dict[str, List[List[float]]],
 9      nu_query: float,
10      method: Literal["linear", "quadratic", "cubic"] = "linear"
11  ) -> Dict[str, Any]:
12      """
13      Interpolate OpInf operators to a new parameter value.
14
15      Args:
16          nu_train: Training parameter values (e.g., [0.1, 0.5, 2.0])
17          operators_train: Dict of operator matrices at each nu_train
18                          Format: {"A": [matrix_at_nu1, matrix_at_nu2, ...],
19                                    "B": [...], "C": [...]}
20          nu_query: Target parameter value
21          method: Interpolation method (linear, quadratic, cubic)
22
23      Returns:
24          Dict with interpolated operators and metadata
25      """
26      nu_array = np.array(nu_train)
27      operator_names = list(operators_train.keys())
28
29      # Validate method against number of points
30      if method == "quadratic" and len(nu_train) < 3:
```

```
31          method = "linear"
32      if method == "cubic" and len(nu_train) < 4:
33          method = "quadratic" if len(nu_train) >= 3 else "linear"
34
35      interpolated = {}
36
37      for op_name in operator_names:
38          # Stack operator matrices
39          op_stack = np.array(operators_train[op_name])  # (n_params, ...)
40
41          # Interpolate element-wise
42          interp_func = interp1d(
43              nu_array, op_stack, axis=0,
44              kind=method,
45              fill_value='extrapolate'
46          )
47
48          op_interp = interp_func(nu_query)
49
50          interpolated[op_name] = {
51              "values": op_interp.tolist(),
52              "shape": list(op_interp.shape),
53              "norm": float(np.linalg.norm(op_interp)),
54              "mean": float(np.mean(op_interp)),
55              "std": float(np.std(op_interp)),
56              "min": float(np.min(op_interp)),
57              "max": float(np.max(op_interp))
58          }
59
60      return {
61          "operators": interpolated,
62          "method": method,
63          "nu_query": nu_query,
64          "success": True
65      }
66
67
68  def linear_regress_operators(
69      nu_train: List[float],
70      operators_train: Dict[str, List[List[float]]],
71      nu_query: float,
72  ) -> Dict[str, Any]:
73      """
74      Linear regression of operators vs normalized nu.
75      Fits y = a*z + b per entry, where z = (nu - mean) / std.
76      """
77      nu_array = np.array(nu_train, dtype=float)
78      nu_mean = float(nu_array.mean())
79      nu_std = float(nu_array.std()) if float(nu_array.std()) > 1e-12 else 1.0
80      z = (nu_array - nu_mean) / nu_std
81      z_q = (float(nu_query) - nu_mean) / nu_std
82
83      operator_names = list(operators_train.keys())
84      outputs = {}
85      for op_name in operator_names:
86          op_stack = np.array(operators_train[op_name])  # (n_params, ...)
87          flat = op_stack.reshape(len(nu_array), -1)
88          X = np.vstack([z, np.ones_like(z)]).T
89          coeffs, _, _, _ = np.linalg.lstsq(X, flat, rcond=None)
90          a = coeffs[0]
91          b = coeffs[1]
92          pred_flat = a * z_q + b
93          pred = pred_flat.reshape(op_stack.shape[1:])
94          outputs[op_name] = {
95              "values": pred.tolist(),
96              "shape": list(pred.shape),
97              "norm": float(np.linalg.norm(pred)),
98              "mean": float(np.mean(pred)),
```

```
99              "std": float(np.std(pred)),
100             "min": float(np.min(pred)),
101             "max": float(np.max(pred)),
102         }
103
104     return {
105         "operators": outputs,
106         "method": "regression",
107         "nu_query": nu_query,
108         "success": True,
109     }
110
111
112 def linear_regress_operators_batch(
113     nu_train: List[float],
114     operators_train: Dict[str, List[List[float]]],
115     nu_queries: List[float],
116 ) -> Dict[str, Any]:
117     """Batch linear regression for multiple nu values."""
118     outputs = {}
119     for nu_query in nu_queries:
120         outputs[str(nu_query)] = linear_regress_operators(nu_train, operators_train,
        nu_query)
121     return {
122         "nu_queries": nu_queries,
123         "method": "regression",
124         "predictions": outputs,
125         "success": True,
126     }
127
128
129 def analyze_parameter_range(
130     nu_train: List[float],
131     nu_query: float
132 ) -> Dict[str, Any]:
133     """
134     Analyze whether query is interpolation or extrapolation.
135
136     Args:
137         nu_train: Training parameter values
138         nu_query: Query parameter value
139
140     Returns:
141         Analysis of parameter range
142     """
143     nu_min = min(nu_train)
144     nu_max = max(nu_train)
145
146     is_interpolation = nu_min <= nu_query <= nu_max
147
148     # Calculate relative position
149     if is_interpolation:
150         if nu_max != nu_min:
151             relative_position = (nu_query - nu_min) / (nu_max - nu_min)
152         else:
153             relative_position = 0.5
154     else:
155         relative_position = None
156
157     # Determine extrapolation distance
158     if nu_query < nu_min:
159         extrapolation_distance = abs(nu_query - nu_min)
160         extrapolation_direction = "below"
161     elif nu_query > nu_max:
162         extrapolation_distance = abs(nu_query - nu_max)
163         extrapolation_direction = "above"
164     else:
165         extrapolation_distance = 0
```

```
166             extrapolation_direction = None
167
168         return {
169             "nu_train": nu_train,
170             "nu_query": nu_query,
171             "nu_range": [nu_min, nu_max],
172             "is_interpolation": is_interpolation,
173             "relative_position": relative_position,
174             "extrapolation_distance": extrapolation_distance,
175             "extrapolation_direction": extrapolation_direction,
176             "confidence": "high" if is_interpolation else "low",
177             "recommendation": "Use linear interpolation" if is_interpolation else
178                             "Extrapolation detected - results may be less accurate"
179         }
180
181
182     def validate_operators(
183         operators: Dict[str, Dict],
184         equation_type: str = "heat"
185     ) -> Dict[str, Any]:
186         """
187         Validate physical constraints on operators.
188
189         Args:
190             operators: Interpolated operators with 'values' field
191             equation_type: Type of equation (heat, burgers, etc.)
192
193         Returns:
194             Validation results
195         """
196         validations = {}
197
198         for op_name, op_data in operators.items():
199             op_array = np.array(op_data["values"])
200
201             checks = {
202                 "has_nan": bool(np.any(np.isnan(op_array))),
203                 "has_inf": bool(np.any(np.isinf(op_array))),
204                 "is_finite": bool(np.all(np.isfinite(op_array))),
205                 "max_abs_value": float(np.max(np.abs(op_array)))
206             }
207
208             # Equation-specific checks
209             if equation_type == "heat":
210                 # For heat equation, A should have negative eigenvalues (stable)
211                 if op_name == "A" and len(op_array.shape) == 2 and op_array.shape[0] == op_array
        .shape[1]:
212                     try:
213                         eigvals = np.linalg.eigvals(op_array)
214                         checks["max_real_eigenvalue"] = float(np.max(np.real(eigvals)))
215                         checks["is_stable"] = checks["max_real_eigenvalue"] < 0
216                     except:
217                         checks["eigenvalue_check"] = "failed"
218
219             elif equation_type == "burgers":
220                 # For Burgers equation, check H tensor shape
221                 if op_name == "H":
222                     checks["shape_info"] = f"H tensor: {op_array.shape}"
223
224             validations[op_name] = checks
225
226         # Overall validation
227         all_valid = all(
228             not v["has_nan"] and not v["has_inf"] and v["is_finite"]
229             for v in validations.values()
230         )
231
232         return {
```

Table 5: Performance of OpInf-LLM under direct code generation without external tools.

| Model | Success Rate | Heat Attempt | Heat Error | Burgers Attempt | Burgers Error | Cavity Attempt | Cavity Error |
|---|---|---|---|---|---|---|---|
| Gemini-2.0 | 99.2% | 1 | 1.29e$-$2 | 1 | 4.91e$-$1 | 2 | 3.88e$-$1 |
| GPT-4o | 99.2% | 1 | 1.87e$-$2 | 4 | 4.91e$-$1 | 5 | 3.26e$-$2 |
| GPT-4.1 | 99.2% | 1 | 1.29e$-$2 | 1 | 4.91e$-$1 | 1 | 3.40e$-$2 |

```
233        "is_valid": all_valid,
234        "operator_checks": validations,
235        "equation_type": equation_type
236    }
```

For code generation, we prompt the LLM to directly generate code for polynomial regression or interpolation to predict reduced operators, as well as an ODE integrator to solve the resulting ROM. The corresponding results across multiple LLM backends are summarized in Table 5. Overall, the proposed OpInf-LLM achieves high success rates, with executable code obtained within five attempts for all tested models and PDEs. The resulting prediction errors are comparable to those of the more deterministic tool-calling pipeline, highlighting the flexibility and effectiveness of OpInf-LLM for reduced-order PDE solving across different LLM execution modes.

Prompts used for code generation are shown below.

**Prompts:**

```
Heat/Burgers':

COEFFICIENT JSON STRUCTURE (IMPORTANT):
- JSON file at: {coeff_path}
- Top-level keys: equation, n_modes, pod_basis_shape, parameters, pod_basis
- parameters is a list; each item has:
  * nu (float)
  * operators: {{A,B,C}} for heat or {{H,A,B,C}} for burgers
  * operators.<name>.values holds the array
- pod_basis.values is the POD basis (space x r) to use as phi

When use_json=True:
- Load operators from the JSON, NOT from per_nu_models in the PKL.
- Use phi from coeff JSON (pod_basis.values).
- Use the PKL model ONLY to read x_grid/x_fine for dx.
"""

    return f"""You are an expert Python programmer.

Generate Python code to:
1) Load model from: {model_path}
2) Load case data from: {data_path}
3) Compute operators for the query parameter using method={method}
4) Integrate the ROM and save output to: {output_path}

USE_JSON={str(use_json)}
{coeff_block}

MODEL STRUCTURE (IMPORTANT):
- pickle with keys: per_nu_models (list of dicts), phi, x_grid or x_fine, t_eval
- Each per_nu_models item has keys: "nu", "A", "B", "C" and for Burgers also "H"
- There is NO nested "operators" key. Operators are top-level per entry.
- Operator shapes (must match exactly): {op_shapes}
```

```
- phi shape: {phi_shape} (space x r)

CASE DATA (.npz) contains:
- Y_ref (may be saved as time x space); if Y_ref.shape[0] != phi.shape[0],
    transpose.
- U_ref (may be time x n_inputs); ensure U_ref is (n_inputs x time).
- t_eval (time grid)
- nu (float)

REQUIREMENTS:
- Always read inputs via: case_data = np.load(data_path)
- Always save outputs via this exact call (do not hard-code filenames):
  np.savez(output_path, Y_ref=Y_ref, Y_rom=Y_rom, t_eval=t_eval, nu=nu_query)
- Do NOT hard-code any filenames or paths.
- OUTPUT PATH RULE:
  The runner sets output_path per case/trajectory; you must use output_path
      directly.
  Example only (do NOT hard-code):
    output_path = "codegen/gpt-4o/burgers/regression/llm_codegen_burgers_nu0.07
        _traj12_raw.npz"
- If nu matches a training nu exactly, use that operator without regression.
- For method=regression: per-entry linear regression y = a*nu + b across
    training nus.
  Use numpy only (np.linalg.lstsq or closed form). Do NOT use sklearn.
- Regression implementation (stable): flatten operator to shape (n_train, n_flat
    ),
  build X = [nu_train, ones], solve X @ coeffs = op_flat using lstsq.
  Then op_query_flat = coeffs[0]*nu_query + coeffs[1], reshape to op_shape.
- For method=interpolation: linear interpolation per entry.
  Use flatten-interp-reshape for arrays (np.interp expects 1D values).
- Heat ROM: a' = C + A a + B*u (B is (r,), u is scalar)
- Burgers ROM: a' = C + A a + H(a,a) + B @ u (B is (r x 3), u is 3-vector)
- For H(a,a), use: quad = np.einsum('ijk,j,k->i', H, a, a)
- Use solve_ivp(method='BDF', vectorized=False), fallback to RK45; for burgers,
    if needed fallback to LSODA.
- Always use t_eval from CASE DATA (do not use model t_eval).
- Projection: a0 = phi.T @ (Y_ref[:,0] * dx), after ensuring Y_ref is (space x
    time).
- Ensure a is a 1D vector of length r (e.g., a = np.asarray(a).ravel()) and
    rom_rhs returns shape (r,).
- In rom_rhs: start with a = np.asarray(a).ravel(); if a.size != r, set a = a[:r
    ]; never reshape to (r,1) or (r,r).
- After computing a0, if a0.size != r, set a0 = a0[:r].
- Enforce r = phi.shape[1]; assert A.shape[0] == r and A.shape[1] == r before
    integration.
- Do NOT reference `model` inside helper functions. If a helper needs x_grid/dx/
    phi, pass it as an argument.
- After solve_ivp, set a_rom = sol.y; if a_rom.shape[0] != r, set a_rom = a_rom.
    T.
- Lift: Y_rom = phi @ a_rom.
- Save output with EXACT statement:
  np.savez(output_path, Y_ref=Y_ref, Y_rom=Y_rom, t_eval=t_eval, nu=nu_query)
- Do not print large arrays.
- Heat-specific shapes: B is (r,), do NOT reshape it to (r,r).
- Burgers-specific shapes: B is (r,3). u = [w1(t), w2(t), w3(t)].
- If Y_ref time length != len(t_eval), resample each spatial row to t_eval using
      np.interp.
- U_ref handling:
  * If U_ref.ndim == 1, reshape to (1, -1).
```

```
    * If U_ref.shape[0] == len(t_eval) and U_ref.shape[1] != len(t_eval),
        transpose.
    * After reshape, enforce U_ref.shape[1] == len(t_eval). If not, resample U_ref
        to t_eval
      using np.interp with a linearly spaced time grid over [t_eval[0], t_eval
          [-1]].
    * For heat, use u = float(np.interp(t, t_eval, U_ref[0])).
    * For burgers, build u vector by interp each row; ensure u has shape (3,).

SANITY CHECKS (must pass before saving):
- assert Y_ref.shape[1] == len(t_eval)
- assert Y_rom.shape == Y_ref.shape
- In rom_rhs, return a 1D array (shape (r,)); do NOT return (r,1) or (r,r).

CODE TEMPLATE (use this structure, adjust details):
1) Load model, extract nu_train and operator arrays per entry.
2) For each operator tensor:
   flat = arr.reshape(-1); stack across nu; interp/regress each entry; reshape
       to op_shapes.
3) Load Y_ref/U_ref; if Y_ref.shape[0] != phi.shape[0] or Y_ref.shape[0] == len(
   t_eval), transpose.
   Ensure U_ref is (n_inputs x time).
4) Integrate ROM; save output.

Return only the complete Python code (no explanations).
"""

Cavity:

Generate Python code to:
1) Load cavity model from: {model_path}
2) Load case data from: {data_path}
3) Compute operators for the query Re using method={method}
4) Integrate ROM with RK4 and save output to: {output_path}

MODEL STRUCTURE (IMPORTANT):
- pickle with keys: per_Re_models (list of dicts), phi, x, y, t_eval
- Each per_Re_models item has keys: "Re", "H", "A", "B", "C"
- There is NO nested "operators" key.
- Operator shapes (must match exactly): {op_shapes}
- phi shape: {phi_shape} (state x r)

CASE DATA (.npz) contains:
- Y_omega (n x time)
- Y_psi (n x time)
- U_lid (time)
- t_eval (time grid)
- Re (float)

REQUIREMENTS:
- Always read inputs via: case_data = np.load(data_path)
- Always save outputs via this exact call (do not hard-code filenames):
  np.savez(output_path, Y_omega_fom=Y_omega, Y_psi_fom=Y_psi,
           Y_omega_rom=Y_omega_rom, Y_psi_rom=Y_psi_rom,
           U_lid=U_lid, x=x, y=y, t_eval=t_eval, Re=Re_query)
- Do NOT hard-code any filenames or paths.
- OUTPUT PATH RULE:
```

```
   The runner sets output_path per case/trajectory; you must use output_path
      directly.
   Example only (do NOT hard-code):
     output_path = "codegen/gpt-4o/cavity/regression/llm_codegen_cavity_Re120.0
        _traj2_raw.npz"
- If Re matches a training Re exactly, use that operator without regression.
- For method=regression: per-entry linear regression y = a*Re + b.
  Use numpy only (np.linalg.lstsq or closed form). Do NOT use sklearn.
- Regression implementation (stable): flatten operator to shape (n_train, n_flat
   ),
  build X = [Re_train, ones], solve X @ coeffs = op_flat using lstsq.
  Then op_query_flat = coeffs[0]*Re_query + coeffs[1], reshape to op_shape.
- For method=interpolation: linear interpolation per entry (flatten-interp-
   reshape).
- Modal state dimension is r = phi.shape[1]. Integrate a(t) in R^r.
- Use operators H, A, B, C with quadratic term H(a,a) via einsum:
  quad = np.einsum('ijk,j,k->i', H, a, a)
- Build Y_fom by stacking omega and psi.
- Project with phi.T @ (Y_fom * dA) where dA = dx*dx.
- Integrate with fixed-step RK4 using dt from t_eval.
- Ensure Y_omega, Y_psi are (n x time) and time length == len(t_eval); do NOT
    downsample.
- If Y_omega/Y_psi time length != len(t_eval), resample each spatial row to
   t_eval using np.interp.
- Save Y_omega_rom/Y_psi_rom with the same shape as Y_omega/Y_psi.
- Save output with EXACT statement:
  np.savez(output_path, Y_omega_fom=Y_omega, Y_psi_fom=Y_psi,
           Y_omega_rom=Y_omega_rom, Y_psi_rom=Y_psi_rom,
           U_lid=U_lid, x=x, y=y, t_eval=t_eval, Re=Re_query)
- Do not print large arrays.
- Use op_shapes exactly; do not reshape to other sizes.
- U_lid handling: if U_lid is not length len(t_eval), resample to t_eval using
   np.interp
  on a linear time grid over [t_eval[0], t_eval[-1]].
- Ensure a is 1D (r,) throughout; do not use column vectors.

SANITY CHECKS (must pass before saving):
- assert Y_omega_rom.shape == Y_omega.shape
- assert Y_psi_rom.shape == Y_psi.shape

CODE TEMPLATE (use this structure, adjust details):
1) Load model, extract Re_train and operator arrays per entry.
2) For each operator tensor:
   flat = arr.reshape(-1); stack across Re; interp/regress each entry; reshape
      to op_shapes.
3) Load Y_omega/Y_psi; build Y_fom; project with phi to get a0 (r x 1).
4) Integrate a(t) with RK4 in r-dim; then Y_rom = phi @ a_traj.
5) Split Y_rom into omega/psi using n = Y_omega.shape[0].
6) Save outputs.

Return only the complete Python code (no explanations).
"""
```

## C   Additional Results

### C.1   Ablation on Natural Language Parsing Diversity

In this section, we conduct ablation experiments to demonstrate that the LLM is a critical component for robustly interpreting diverse and unstructured natural language descriptions of PDE problems, compared to traditional structured parsers.

Structured parsers are inherently limited in their ability to handle flexible, non-technical, and imperfect user inputs. In contrast, the LLM enables robust interpretation across a wide range of linguistic variations, including typos, informal phrasing, and implicit physical descriptions.

We evaluate OpInf-LLM against a static parser on 50 natural language scenarios describing PDEs and their parameter settings. The quantitative results are summarized in Table 4. OpInf-LLM achieves perfect accuracy across all metrics, while the static parser shows substantially lower performance. The static parser frequently fails when handling:

- **Implicit descriptions:** e.g., "temperature moving through a metal rod in a cooling line" (heat equation).

- **Informal phrasing:** e.g., "traffic-wave style transport" (Burgers-type dynamics).

- **Typos and casing variations:** e.g., "hEat and burGERs".

- **Composite or underspecified requests:** e.g., "cavity + heat. Re values 100, 150. heat nus 0.1 0.9", where multiple PDEs are combined with implicit parameter associations.

More example failure cases for the static parser are shown below.

```
Let's do hEat and burGERs (typos intentional) and dump in runs/noisy_case

I want to study how temperature moves through a metal rod in a cooling line,
with diffusion rates 0.1 and 0.5.

Run the traffic-wave style transport case where sharp fronts develop,
at viscosity 0.03 and 0.07.

cavity + heat. Re values 100, 150. heat nus 0.1 0.9

Use Fourier-like conduction case plus the nonlinear convective transport one
(Burgers-style); skip cavity.

For a room-air circulation benchmark in a square box with a moving top boundary,
test Reynolds 80 and 120.

Please do conduction (heat-like) + cavity flow,
place outputs in ablations/implicit_phys_33, raw true.

I care only about the square-cavity moving-lid flow benchmark;
Reynolds sweep 60 80 90 110 120 140.
output folder experiments/re_sweep.

Please run a one-dimensional thermal diffusion check for process control tuning
at 0.5 and 3.0.
```

In contrast, OpInf-LLM consistently maps these inputs to the correct PDE types and parameter configurations. For example, it correctly interprets *"temperature moving through a metal rod in a cooling line with diffusion rates 0.1 and 0.5"* as a heat equation with the corresponding diffusion coefficients. These results demonstrate

Table 6: POD basis ablation for OpInf-LLM.

| Equation | POD | Energy (%) | Error $[0, T]$ | Error $[T, 2T]$ |
|---|---|---|---|---|
| Burgers | 4 | 88.01 | 3.93e−1 | 4.99e−1 |
| Burgers | 5 | 92.26 | 3.44e−1 | 4.48e−1 |
| Burgers | 6 | 95.63 | 4.13e−1 | 9.63e−1 |
| Burgers | 7 | 97.26 | 3.52e−1 | 5.41e−1 |
| Burgers | 8 | 98.31 | 3.64e−1 | 4.90e−1 |
| Burgers | 9 | 98.85 | 4.70e−1 | 6.46e−1 |
| Burgers | 10 | 99.20 | 4.91e−1 | 6.36e−1 |

Table 7: Regularization intensity ($\lambda$) ablation for OpInf-LLM.

| Equation | $\lambda$ | Norm | Error $[0, T]$ | Error $[T, 2T]$ |
|---|---|---|---|---|
| Burgers | 0 | 1730.53 | 2.25e1 | 2.92e1 |
| Burgers | $10^{-6}$ | 1730.53 | 2.25e1 | 2.92e1 |
| Burgers | $10^{-5}$ | 1645.36 | 2.25e1 | 2.92e1 |
| Burgers | $10^{-4}$ | 1645.28 | 2.25e1 | 2.92e1 |
| Burgers | $10^{-3}$ | 1637.70 | 2.26e1 | 2.94e1 |
| Burgers | $10^{-2}$ | 1377.88 | 2.96e1 | 3.90e1 |
| Burgers | $10^{-1}$ | 346.36 | 1.81e1 | 5.42e1 |
| Burgers | 1 | 76.14 | 3.31e−1 | 5.53e−1 |
| Burgers | 10 | 33.22 | 3.34e−1 | 4.53e−1 |

that the LLM is not merely a convenience layer, but a key component that enables robust and flexible problem specification, including the ability to interpret non-technical and implicitly defined descriptions. This significantly improves reliability under realistic user inputs.

## C.2 Number of POD Modes Ablations

We examine the effect of the number of POD basis modes in the proposed OpInf-LLM method. Specifically, we follow the settings in Section 4.1 and vary the number of POD modes $r$ from 4 to 10 for each equation instance. We report the total energy captured by the POD basis on the training data, along with the prediction errors on test parameters and configurations for both the original and extended time horizons. Table 6 summarizes the results for Burgers' equation with full results available in Table 11 in the Appendix. As expected, increasing the number of POD modes captures more energy in the training data and generally improves prediction accuracy, provided that the additional modes are not overfitting. In practice, capturing around 99% of the energy serves as a useful guideline for selecting the number of POD modes (Kramer et al., 2024). Overly large POD bases can lead to solver instability, as observed for the heat equation with a large number of modes. Nevertheless, the OpInf-LLM framework remains executable across these settings with 100% code success rate, indicating robustness with respect to the choice of POD dimension.

## C.3 Regularization Intensity Ablations

We examine the effect of the regularization strength $\lambda$ in equation 8 on the proposed OpInf-LLM scheme. Following the settings in Section 4.1, we vary $\lambda$ logarithmically from 0 to 10 for each equation instance, and report the averaged operator norm $\sqrt{\|A\|_F^2 + \|H\|_F^2 + \|B\|_F^2 + \|c\|_F^2}$ on the training set, together with the prediction errors on test configurations for both the original and extended time horizons. Table 7 shows the results for Burgers' equation and full results are available in Table 12 in the Appendix. The averaged operator norms decrease as the regularization strength increases, and the best $\lambda$ for test performance can be identified, which is typically within a range spanning one order of magnitude or more. However, certain regularization values (e.g., $\lambda = 0$ and $10^{-6}$ for the cavity flow) can lead to failed least-squares optimization due to numerical singularity. In practice, ablations over a range of regularization values are typically performed on a validation

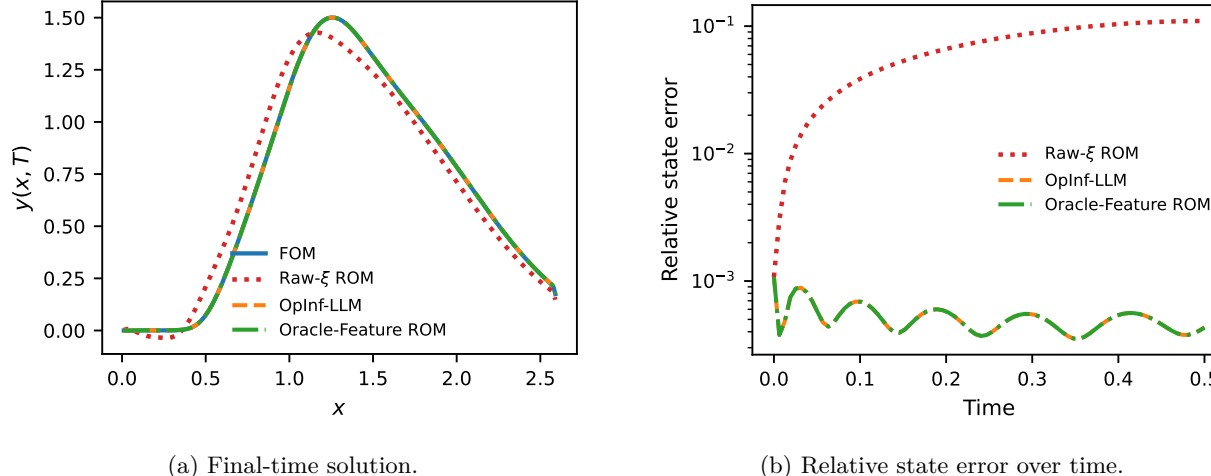

(a) Final-time solution.

(b) Relative state error over time.

Figure 6: Prediction comparison for the feature reasoning example.

Table 8: Relative trajectory errors for the parametric ROM feature-identification test. The raw-$\xi$ ROM uses the naive feature map $\{1, \xi\}$, while the OpInf-LLM ROM uses the feature map identified from the neutral prompt. The OpInf-LLM ROM recovers the oracle feature map and substantially improves test accuracy.

| Test parameter $\xi$ | Raw-$\xi$ ROM | OpInf-LLM ROM | Oracle ROM |
|---|---|---|---|
| 0.75 | $3.0119 \times 10^{-2}$ | $4.5243 \times 10^{-4}$ | $4.5243 \times 10^{-4}$ |
| 1.60 | $7.7841 \times 10^{-2}$ | $5.2922 \times 10^{-4}$ | $5.2922 \times 10^{-4}$ |
| 2.00 | $2.8715 \times 10^{0}$ | $5.8492 \times 10^{-4}$ | $5.8492 \times 10^{-4}$ |

set, and the operator norm is jointly considered to determine an appropriate $\lambda$, with low validation error and a moderate operator norm being desired (Golub et al., 1979; McQuarrie et al., 2021).

## C.4 Feature Reasoning

We include an additional test to evaluate whether OpInf-LLM can identify nontrivial parameter features for parametric ROM construction without being given the correct feature map.

We consider the one-dimensional advection–diffusion equation

$$y_t + a y_x = \kappa y_{xx}, \qquad x \in [0, L(\xi)], \qquad L(\xi) = 1 + \xi, \tag{31}$$

where $a$ and $\kappa$ are fixed constants and $\xi$ is the parameter. Although $\xi$ does not appear explicitly as a coefficient in the physical-domain PDE, the computational grid for a parametric ROM must be fixed across different parameter values. Mapping the parameter-dependent physical domain to the reference coordinate

$$\bar{x} = \frac{x}{L(\xi)}, \qquad \bar{x} \in [0, 1], \tag{32}$$

reveals the actual parameter dependence of the operators on the fixed grid, where the first-derivative operator scales with $1/L(\xi)$, while the second-derivative operator scales with $1/L(\xi)^2$. Therefore, the physically consistent parametric OpInf model should use the feature map

$$\left[ \frac{1}{1+\xi}, \frac{1}{(1+\xi)^2} \right], \tag{33}$$

yielding the reduced model structure

$$\dot{a} = \frac{1}{1+\xi} A_1 a + \frac{1}{(1+\xi)^2} A_2 a. \tag{34}$$

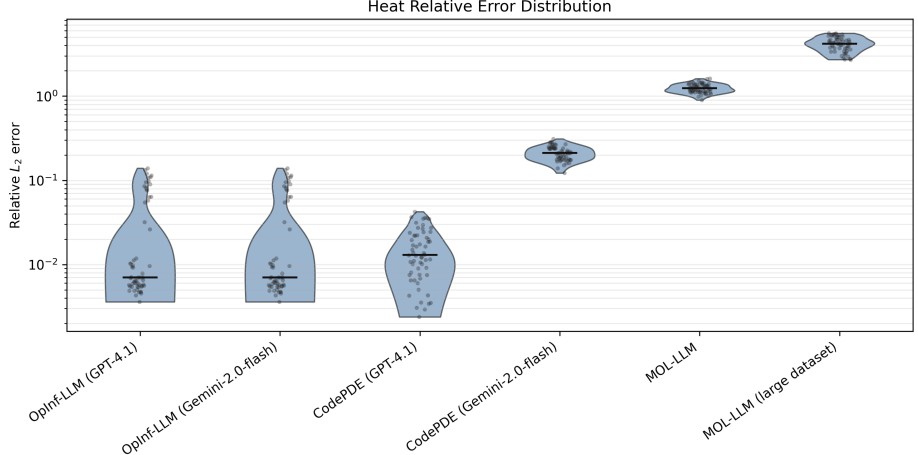

Figure 7: Trajectory-level relative error distributions for the heat equations.

We provide OpInf-LLM only with the physical-domain problem statement in equation 31 and ask: *"What parameter features should be used to parameterize the reduced operators in a parametric OpInf model?"* The prompt does not mention the reference domain transformation, derivative rescaling, or the target feature map.

We then test the prediction performance of OpInf-LLM with gpt-4.1-mini using offline full-order model (FOM) data at $\mu_{\text{train}} = \{0.0, 0.2, 0.5, 0.9, 1.3\}$. The FOM is discretized on the fixed reference domain with $N = 250$ spatial degrees of freedom. The POD basis uses $r = 12$ modes and captures $99.999982\%$ of the snapshot energy.

For comparison, we also test a standard raw-parameter OpInf model

$$\dot{a} = A_0 a + \xi A_\xi a, \tag{35}$$

which assumes an affine dependence on the raw parameter $\xi$ rather than the geometry-induced feature map. We further include an oracle ROM using the analytically derived features in equation 33.

Fig. 6 compares the predicted solution and the time-dependent relative state error for the unseen test parameter $\xi = 1.6$. The raw-$\xi$ ROM deviates from the FOM because its affine dependence on $\xi$ does not match the actual operator dependence on the fixed reference grid. In contrast, OpInf-LLM correctly infers the feature map in equation 33, which agrees with the oracle ROM and closely tracks the FOM.

Table 8 reports the relative trajectory error on unseen test parameters, computed over the full sampled space–time trajectory. The raw-$\xi$ ROM gives substantially larger errors, and its performance deteriorates for the extrapolatory test case $\xi = 2.0$. In contrast, OpInf-LLM matches the oracle ROM, showing that the LLM correctly identifies the relevant parametric OpInf features from the neutral problem description.

### C.5   Comparison with Parametric OpInf

We additionally compare OpInf-LLM with a manually constructed parametric OpInf reference. This comparison is included for reference rather than as a direct baseline, since the parametric OpInf reference does not take natural-language instructions as input and requires manual configuration and implementation. Specifically, with parametric OpInf, we manually determine the governing equation, identify the appropriate parameter-dependent structure, construct the reduced regression features, and implement the parametric OpInf procedure. In contrast, OpInf-LLM is given a natural-language problem description and uses the OpInf formulation to automatically generate the corresponding reduced model construction and executable code.

The prediction results are reported in Table 9. The manually constructed parametric OpInf reference achieves the same errors as the tool-based OpInf-LLM implementations across all tested LLM backends. This

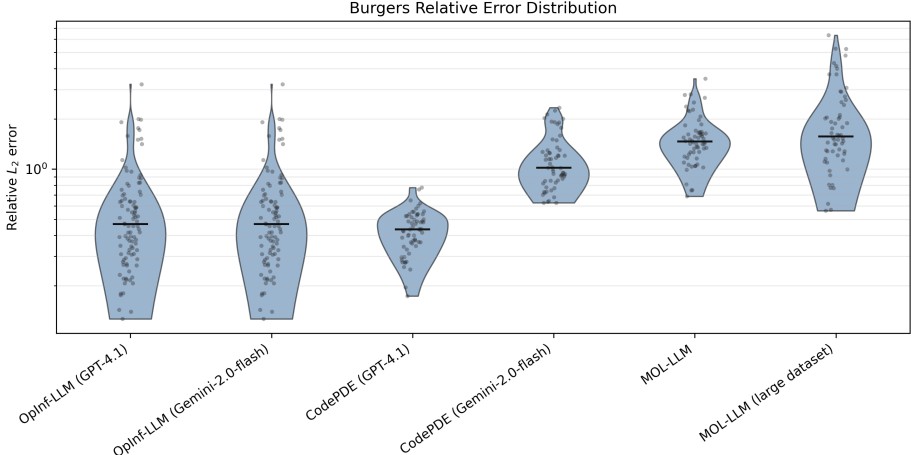

Figure 8: Trajectory-level relative error distributions for the Burgers' equations.

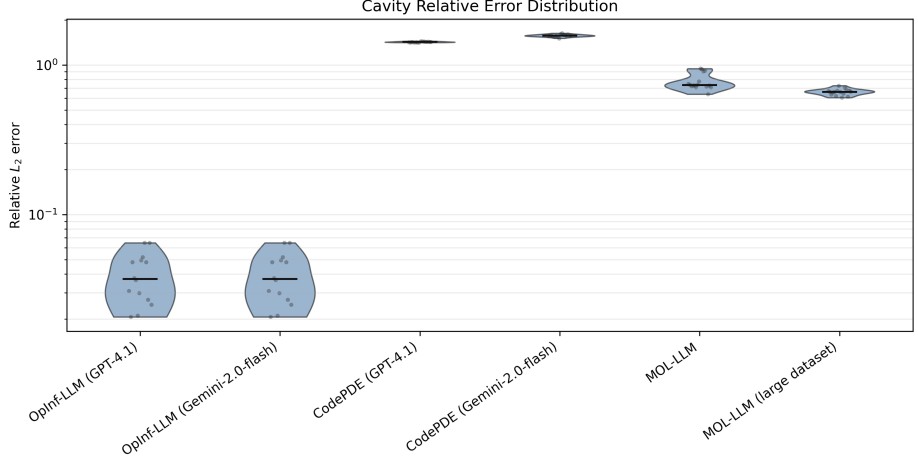

Figure 9: Trajectory-level relative error distributions for the lid-driven cavity equations.

agreement is expected, because once the correct parametric OpInf structure and feature map are specified, both approaches solve the same reduced regression problem using the same data and model dimension. The code-generation version achieves comparable results, with minor differences attributable mainly to implementation variations in feature identification and ODE integration.

This comparison helps isolate the role of the LLM component and the structured OpInf formulation. The purpose of OpInf-LLM is not to improve the numerical regression accuracy beyond a correctly specified parametric OpInf model constructed by a human expert. Rather, its contribution is to make the language-guided scientific computing task more structured and reliable. Instead of asking the LLM to construct a full PDE solver from scratch, OpInf-LLM guides the model to identify the equation-specific operator structure, parameter dependence, and regression features needed for a reduced OpInf representation. This reduces the difficulty of the generation task and improves consistency across different equations and LLM backends.

### C.6   Trajectory-Level Error Distributions

In this section, we report the trajectory-level relative $L^2$ error distributions on the test set for the experiments in Section 4. For the proposed OpInf-LLM, we report results from its tool-use version. Fig. 7–9 show the error distributions for the heat, Burgers, and lid-driven cavity problems, respectively. Each data point represents

Table 9: Comparison with manually constructed parametric OpInf reference.

| Method | Data | Train time (s) | Success rate ↑ | Heat ↓ | Burgers ↓ | Cavity ↓ |
|---|---|---|---|---|---|---|
| Parametric OpInf reference | 449 | 30 | 99.2% | 1.29e−2 | 4.91e−1 | 4.63e−2 |
| OpInf-LLM (tool, GPT-4.1) | 449 | 30 | 99.2% | 1.29e−2 | 4.91e−1 | 4.63e−2 |
| OpInf-LLM (tool, GPT-4o) | 449 | 30 | 99.2% | 1.29e−2 | 4.91e−1 | 4.63e−2 |
| OpInf-LLM (tool, Gemini-2.0-flash) | 449 | 30 | 99.2% | 1.29e−2 | 4.91e−1 | 4.63e−2 |
| OpInf-LLM (codegen, GPT-4.1) | 449 | 30 | 99.2% | 1.29e−2 | 4.91e−1 | 3.40e−2 |
| OpInf-LLM (codegen, GPT-4o) | 449 | 30 | 99.2% | 1.87e−2 | 4.91e−1 | 3.26e−2 |
| OpInf-LLM (codegen, Gemini-2.0-flash) | 449 | 30 | 99.2% | 1.29e−2 | 4.91e−1 | 3.88e−1 |

the relative error of one test trajectory under a specific test parameter setting, and the displayed distributions aggregate errors across all test trajectories and all parametric test settings. It can be observed that the proposed OpInf-LLM pipeline generally achieves lower trajectory-level errors across the three problems compared with CodePDE, MOL-LLM, and their variants, while exhibiting more consistent performance across the tested LLM backbones than CodePDE.

## C.7 Full Tables and Visualizations

In this section, we provide full experiment and ablation results. While we consider polynomial regression for predicting reduced operators in the main draft, we additionally consider interpolation methods. Performance for both methods are comparable and the quantitative results are shown in Table 10, with errors in extended time horizon reported. Full ablation results on the number of POD basis and regularization intensity are shown in Table 11 and Table 12, respectively. Additional visualizations are shown in Fig. 10–13.

Table 10: OpInf-LLM full results.

| LLM model | Equation | Method | Parameter | Error $[0, T]$ | Error $[T, 2T]$ | Success rate (%) |
|---|---|---|---|---|---|---|
| GPT-4.1 | heat | interpolation | 0.5 | 8.70e−3 | 5.97e−3 | 100 |
| GPT-4.1 | heat | interpolation | 1.0 | 6.07e−3 | 5.24e−3 | 100 |
| GPT-4.1 | heat | interpolation | 3.0 | 2.22e−2 | 1.17e−1 | 100 |
| GPT-4.1 | heat | regression | 0.5 | 8.70e−3 | 5.97e−3 | 100 |
| GPT-4.1 | heat | regression | 1.0 | 1.02e−2 | 3.08e−2 | 100 |
| GPT-4.1 | heat | regression | 3.0 | 1.99e−2 | 7.12e−2 | 100 |
| GPT-4.1 | burgers | interpolation | 0.03 | 4.81e−1 | 5.96e−1 | 95 |
| GPT-4.1 | burgers | interpolation | 0.07 | 4.33e−1 | 4.97e−1 | 100 |
| GPT-4.1 | burgers | regression | 0.03 | 4.50e−1 | 6.56e−1 | 95 |
| GPT-4.1 | burgers | regression | 0.07 | 5.32e−1 | 6.15e−1 | 100 |
| GPT-4.1 | cavity | interpolation | 60 | 5.74e−2 | 5.94e−2 | 100 |
| GPT-4.1 | cavity | interpolation | 80 | 3.95e−2 | 3.54e−2 | 100 |
| GPT-4.1 | cavity | interpolation | 90 | 4.76e−2 | 4.15e−2 | 100 |
| GPT-4.1 | cavity | interpolation | 110 | 2.51e−2 | 2.64e−2 | 100 |
| GPT-4.1 | cavity | interpolation | 120 | 4.23e−2 | 4.34e−2 | 100 |
| GPT-4.1 | cavity | interpolation | 140 | 2.92e−2 | 2.83e−2 | 100 |
| GPT-4.1 | cavity | regression | 60 | 5.54e−2 | 5.57e−2 | 100 |
| GPT-4.1 | cavity | regression | 80 | 4.87e−2 | 5.13e−2 | 100 |
| GPT-4.1 | cavity | regression | 90 | 5.83e−2 | 5.94e−2 | 100 |
| GPT-4.1 | cavity | regression | 110 | 3.57e−2 | 4.14e−2 | 100 |
| GPT-4.1 | cavity | regression | 120 | 4.44e−2 | 4.60e−2 | 100 |
| GPT-4.1 | cavity | regression | 140 | 3.50e−2 | 3.78e−2 | 100 |

*Continued on next page*

| LLM model | Equation | Method | Parameter | Error $[0, T]$ | Error $[T, 2T]$ | Success rate (%) |
|---|---|---|---|---|---|---|
| GPT-4o | heat | interpolation | 0.5 | 8.70e−3 | 5.97e−3 | 100 |
| GPT-4o | heat | interpolation | 1.0 | 6.07e−3 | 5.24e−3 | 100 |
| GPT-4o | heat | interpolation | 3.0 | 2.22e−2 | 1.17e−1 | 100 |
| GPT-4o | heat | regression | 0.5 | 8.70e−3 | 5.97e−3 | 100 |
| GPT-4o | heat | regression | 1.0 | 1.02e−2 | 3.08e−2 | 100 |
| GPT-4o | heat | regression | 3.0 | 1.99e−2 | 7.12e−2 | 100 |
| GPT-4o | burgers | interpolation | 0.03 | 4.81e−1 | 5.96e−1 | 95 |
| GPT-4o | burgers | interpolation | 0.07 | 4.33e−1 | 4.97e−1 | 100 |
| GPT-4o | burgers | regression | 0.03 | 4.50e−1 | 6.56e−1 | 95 |
| GPT-4o | burgers | regression | 0.07 | 5.32e−1 | 6.15e−1 | 100 |
| GPT-4o | cavity | interpolation | 60 | 5.74e−2 | 5.94e−2 | 100 |
| GPT-4o | cavity | interpolation | 80 | 3.95e−2 | 3.54e−2 | 100 |
| GPT-4o | cavity | interpolation | 90 | 4.76e−2 | 4.15e−2 | 100 |
| GPT-4o | cavity | interpolation | 110 | 2.51e−2 | 2.64e−2 | 100 |
| GPT-4o | cavity | interpolation | 120 | 4.23e−2 | 4.34e−2 | 100 |
| GPT-4o | cavity | interpolation | 140 | 2.92e−2 | 2.83e−2 | 100 |
| GPT-4o | cavity | regression | 60 | 5.54e−2 | 5.57e−2 | 100 |
| GPT-4o | cavity | regression | 80 | 4.87e−2 | 5.13e−2 | 100 |
| GPT-4o | cavity | regression | 90 | 5.83e−2 | 5.94e−2 | 100 |
| GPT-4o | cavity | regression | 110 | 3.57e−2 | 4.14e−2 | 100 |
| GPT-4o | cavity | regression | 120 | 4.44e−2 | 4.60e−2 | 100 |
| GPT-4o | cavity | regression | 140 | 3.50e−2 | 3.78e−2 | 100 |
| Gemini-2.0-flash | heat | interpolation | 0.5 | 8.70e−3 | 5.97e−3 | 100 |
| Gemini-2.0-flash | heat | interpolation | 1.0 | 6.07e−3 | 5.24e−3 | 100 |
| Gemini-2.0-flash | heat | interpolation | 3.0 | 2.22e−2 | 1.17e−1 | 100 |
| Gemini-2.0-flash | heat | regression | 0.5 | 8.70e−3 | 5.97e−3 | 100 |
| Gemini-2.0-flash | heat | regression | 1.0 | 1.02e−2 | 3.08e−2 | 100 |
| Gemini-2.0-flash | heat | regression | 3.0 | 1.99e−2 | 7.12e−2 | 100 |
| Gemini-2.0-flash | burgers | interpolation | 0.03 | 4.81e−1 | 5.96e−1 | 95 |
| Gemini-2.0-flash | burgers | interpolation | 0.07 | 4.33e−1 | 4.97e−1 | 100 |
| Gemini-2.0-flash | burgers | regression | 0.03 | 4.50e−1 | 6.56e−1 | 95 |
| Gemini-2.0-flash | burgers | regression | 0.07 | 5.32e−1 | 6.15e−1 | 100 |
| Gemini-2.0-flash | cavity | interpolation | 60 | 5.74e−2 | 5.94e−2 | 100 |
| Gemini-2.0-flash | cavity | interpolation | 80 | 3.95e−2 | 3.54e−2 | 100 |
| Gemini-2.0-flash | cavity | interpolation | 90 | 4.76e−2 | 4.15e−2 | 100 |
| Gemini-2.0-flash | cavity | interpolation | 110 | 2.51e−2 | 2.64e−2 | 100 |
| Gemini-2.0-flash | cavity | interpolation | 120 | 4.23e−2 | 4.34e−2 | 100 |
| Gemini-2.0-flash | cavity | interpolation | 140 | 2.92e−2 | 2.83e−2 | 100 |
| Gemini-2.0-flash | cavity | regression | 60 | 5.54e−2 | 5.57e−2 | 100 |
| Gemini-2.0-flash | cavity | regression | 80 | 4.87e−2 | 5.13e−2 | 100 |
| Gemini-2.0-flash | cavity | regression | 90 | 5.83e−2 | 5.94e−2 | 100 |
| Gemini-2.0-flash | cavity | regression | 110 | 3.57e−2 | 4.14e−2 | 100 |
| Gemini-2.0-flash | cavity | regression | 120 | 4.44e−2 | 4.60e−2 | 100 |
| Gemini-2.0-flash | cavity | regression | 140 | 3.50e−2 | 3.78e−2 | 100 |
| Qwen Plus | heat | interpolation | 0.5 | 8.70e−3 | 5.97e−3 | 100 |
| Qwen Plus | heat | interpolation | 1.0 | 6.07e−3 | 5.24e−3 | 100 |
| Qwen Plus | heat | interpolation | 3.0 | 2.22e−2 | 1.17e−1 | 100 |
| Qwen Plus | heat | regression | 0.5 | 8.70e−3 | 5.97e−3 | 100 |

*Continued on next page*

| LLM model | Equation | Method | Parameter | Error $[0, T]$ | Error $[T, 2T]$ | Success rate (%) |
|---|---|---|---|---|---|---|
| Qwen Plus | heat | regression | 1.0 | 1.02e−2 | 3.08e−2 | 100 |
| Qwen Plus | heat | regression | 3.0 | 1.99e−2 | 7.12e−2 | 100 |
| Qwen Plus | burgers | interpolation | 0.03 | 4.81e−1 | 5.96e−1 | 95 |
| Qwen Plus | burgers | interpolation | 0.07 | 4.33e−1 | 4.97e−1 | 100 |
| Qwen Plus | burgers | regression | 0.03 | 4.50e−1 | 6.56e−1 | 95 |
| Qwen Plus | burgers | regression | 0.07 | 5.32e−1 | 6.15e−1 | 100 |
| Qwen Plus | cavity | interpolation | 60 | 6.35e−2 | 6.93e−2 | 100 |
| Qwen Plus | cavity | interpolation | 80 | 3.90e−2 | 3.52e−2 | 100 |
| Qwen Plus | cavity | interpolation | 90 | 4.68e−2 | 4.11e−2 | 100 |
| Qwen Plus | cavity | interpolation | 110 | 2.45e−2 | 2.61e−2 | 100 |
| Qwen Plus | cavity | interpolation | 120 | 4.21e−2 | 4.33e−2 | 100 |
| Qwen Plus | cavity | interpolation | 140 | 2.98e−2 | 3.08e−2 | 100 |
| Qwen Plus | cavity | regression | 60 | 5.54e−2 | 5.57e−2 | 100 |
| Qwen Plus | cavity | regression | 80 | 4.87e−2 | 5.13e−2 | 100 |
| Qwen Plus | cavity | regression | 90 | 5.83e−2 | 5.94e−2 | 100 |
| Qwen Plus | cavity | regression | 110 | 3.57e−2 | 4.14e−2 | 100 |
| Qwen Plus | cavity | regression | 120 | 4.44e−2 | 4.60e−2 | 100 |
| Qwen Plus | cavity | regression | 140 | 3.50e−2 | 3.78e−2 | 100 |
| Claude Sonnet 4 | heat | interpolation | 0.5 | 8.70e−3 | 5.97e−3 | 100 |
| Claude Sonnet 4 | heat | interpolation | 1.0 | 6.07e−3 | 5.24e−3 | 100 |
| Claude Sonnet 4 | heat | interpolation | 3.0 | 2.22e−2 | 1.17e−1 | 100 |
| Claude Sonnet 4 | heat | regression | 0.5 | 8.70e−3 | 5.97e−3 | 100 |
| Claude Sonnet 4 | heat | regression | 1.0 | 1.02e−2 | 3.08e−2 | 100 |
| Claude Sonnet 4 | heat | regression | 3.0 | 1.99e−2 | 7.12e−2 | 100 |
| Claude Sonnet 4 | burgers | interpolation | 0.03 | 4.81e−1 | 5.96e−1 | 95 |
| Claude Sonnet 4 | burgers | interpolation | 0.07 | 4.33e−1 | 4.97e−1 | 100 |
| Claude Sonnet 4 | burgers | regression | 0.03 | 4.50e−1 | 6.56e−1 | 95 |
| Claude Sonnet 4 | burgers | regression | 0.07 | 5.32e−1 | 6.15e−1 | 100 |
| Claude Sonnet 4 | cavity | interpolation | 60 | 6.35e−2 | 6.93e−2 | 100 |
| Claude Sonnet 4 | cavity | interpolation | 80 | 3.90e−2 | 3.52e−2 | 100 |
| Claude Sonnet 4 | cavity | interpolation | 90 | 4.68e−2 | 4.11e−2 | 100 |
| Claude Sonnet 4 | cavity | interpolation | 110 | 2.45e−2 | 2.61e−2 | 100 |
| Claude Sonnet 4 | cavity | interpolation | 120 | 4.21e−2 | 4.33e−2 | 100 |
| Claude Sonnet 4 | cavity | interpolation | 140 | 2.98e−2 | 3.08e−2 | 100 |
| Claude Sonnet 4 | cavity | regression | 60 | 5.54e−2 | 5.57e−2 | 100 |
| Claude Sonnet 4 | cavity | regression | 80 | 4.87e−2 | 5.13e−2 | 100 |
| Claude Sonnet 4 | cavity | regression | 90 | 5.83e−2 | 5.94e−2 | 100 |
| Claude Sonnet 4 | cavity | regression | 110 | 3.57e−2 | 4.14e−2 | 100 |
| Claude Sonnet 4 | cavity | regression | 120 | 4.44e−2 | 4.60e−2 | 100 |
| Claude Sonnet 4 | cavity | regression | 140 | 3.50e−2 | 3.78e−2 | 100 |

Table 11: POD modes ablation for OpInf-LLM.

| Equation | POD | Method | Error $[0, T]$ | Error $[T, 2T]$ | Error $[0, 2T]$ |
|---|---|---|---|---|---|
| Heat | 4 | interpolation | 3.35e−2 | 5.81e−2 | 4.78e−2 |
| Heat | 4 | regression | 1.96e−2 | 2.39e−2 | 2.21e−2 |
| Heat | 5 | interpolation | 1.77e−2 | 3.72e−2 | 2.98e−2 |
| Heat | 5 | regression | 3.32e−2 | 7.87e−2 | 6.03e−2 |
| Heat | 6 | interpolation | 1.23e−2 | 4.27e−2 | 3.16e−2 |
| Heat | 6 | regression | 1.29e−2 | 3.60e−2 | 2.72e−2 |
| Heat | 7 | interpolation | 7.18e−3 | 1.94e−2 | 1.47e−2 |
| Heat | 7 | regression | 2.03e−2 | 6.00e−2 | 4.44e−2 |
| Heat | 8 | interpolation | 1.07e−3 | 9.44e−4 | 1.02e−3 |
| Heat | 8 | regression | unstable | unstable | unstable |
| Heat | 9 | interpolation | 8.72e−4 | 4.52e−4 | 6.74e−4 |
| Heat | 9 | regression | unstable | unstable | unstable |
| Heat | 10 | interpolation | 7.83e−4 | 2.40e−4 | 5.54e−4 |
| Heat | 10 | regression | unstable | unstable | unstable |
| Burgers | 4 | interpolation | 4.09e−1 | 5.31e−1 | 4.58e−1 |
| Burgers | 4 | regression | 3.93e−1 | 4.99e−1 | 4.36e−1 |
| Burgers | 5 | interpolation | 3.72e−1 | 4.83e−1 | 4.15e−1 |
| Burgers | 5 | regression | 3.44e−1 | 4.48e−1 | 3.86e−1 |
| Burgers | 6 | interpolation | 4.15e−1 | 7.89e−1 | 5.95e−1 |
| Burgers | 6 | regression | 4.13e−1 | 9.63e−1 | 6.94e−1 |
| Burgers | 7 | interpolation | 3.27e−1 | 4.81e−1 | 3.94e−1 |
| Burgers | 7 | regression | 3.52e−1 | 5.41e−1 | 4.38e−1 |
| Burgers | 8 | interpolation | 3.63e−1 | 4.54e−1 | 4.08e−1 |
| Burgers | 8 | regression | 3.64e−1 | 4.90e−1 | 4.24e−1 |
| Burgers | 9 | interpolation | 3.95e−1 | 5.02e−1 | 4.44e−1 |
| Burgers | 9 | regression | 4.70e−1 | 6.46e−1 | 5.50e−1 |
| Burgers | 10 | interpolation | 4.57e−1 | 5.47e−1 | 5.04e−1 |
| Burgers | 10 | regression | 4.91e−1 | 6.36e−1 | 5.67e−1 |
| Cavity | 4 | interpolation | 8.24e−2 | 5.73e−2 | 7.16e−2 |
| Cavity | 4 | regression | 8.64e−2 | 6.39e−2 | 7.66e−2 |
| Cavity | 5 | interpolation | 7.40e−2 | 5.29e−2 | 6.49e−2 |
| Cavity | 5 | regression | 8.00e−2 | 6.22e−2 | 7.21e−2 |
| Cavity | 6 | interpolation | 7.27e−2 | 5.29e−2 | 6.40e−2 |
| Cavity | 6 | regression | 7.79e−2 | 6.17e−2 | 7.07e−2 |
| Cavity | 7 | interpolation | 7.12e−2 | 4.92e−2 | 6.18e−2 |
| Cavity | 7 | regression | 7.58e−2 | 5.66e−2 | 6.73e−2 |
| Cavity | 8 | interpolation | 7.08e−2 | 4.83e−2 | 6.12e−2 |
| Cavity | 8 | regression | 7.62e−2 | 5.53e−2 | 6.72e−2 |
| Cavity | 9 | interpolation | 7.08e−2 | 4.61e−2 | 6.05e−2 |
| Cavity | 9 | regression | 7.68e−2 | 5.53e−2 | 6.77e−2 |
| Cavity | 10 | interpolation | 7.35e−2 | 4.93e−2 | 6.34e−2 |
| Cavity | 10 | regression | 7.74e−2 | 5.77e−2 | 6.89e−2 |

Table 12: Regularization intensity ($\lambda$) ablation for OpInf-LLM.

| Equation | $\lambda$ | Method | Error $[0, T]$ | Error $[T, 2T]$ | Error $[0, 2T]$ |
|---|---|---|---|---|---|
| Heat | 0 | interpolation | 1.23e−2 | 4.27e−2 | 3.16e−2 |
| Heat | 0 | regression | 1.29e−2 | 3.60e−2 | 2.72e−2 |
| Heat | $10^{-6}$ | interpolation | 1.23e−2 | 4.27e−2 | 3.16e−2 |
| Heat | $10^{-6}$ | regression | 1.29e−2 | 3.60e−2 | 2.72e−2 |
| Heat | $10^{-5}$ | interpolation | 1.23e−2 | 4.27e−2 | 3.16e−2 |
| Heat | $10^{-5}$ | regression | 1.29e−2 | 3.60e−2 | 2.72e−2 |
| Heat | $10^{-4}$ | interpolation | 1.23e−2 | 4.27e−2 | 3.16e−2 |
| Heat | $10^{-4}$ | regression | 1.29e−2 | 3.60e−2 | 2.72e−2 |
| Heat | $10^{-3}$ | interpolation | 1.23e−2 | 4.27e−2 | 3.16e−2 |
| Heat | $10^{-3}$ | regression | 1.29e−2 | 3.60e−2 | 2.72e−2 |
| Heat | $10^{-2}$ | interpolation | 1.25e−2 | 4.4e−2 | 3.25e−2 |
| Heat | $10^{-2}$ | regression | 1.22e−2 | 3.33e−2 | 2.54e−2 |
| Heat | $10^{-1}$ | interpolation | 4.00e−2 | 4.43e−1 | 3.07e−1 |
| Heat | $10^{-1}$ | regression | 3.44e−2 | 1.09e−1 | 7.97e−2 |
| Heat | 1 | interpolation | 4.37e3 | 4.43e9 | 3.01e9 |
| Heat | 1 | regression | 5.67e−2 | 8.61e−2 | 7.3e−2 |
| Heat | 10 | interpolation | 3.73e−1 | 1.52e0 | 1.12e0 |
| Heat | 10 | regression | 2.71e−1 | 3.16e−1 | 2.96e−1 |
| Burgers | 0 | interpolation | 1.34e0 | 9.17e−1 | 1.14e0 |
| Burgers | 0 | regression | 2.25e1 | 2.92e1 | 2.5e1 |
| Burgers | $10^{-6}$ | interpolation | 1.34e0 | 9.17e−1 | 1.14e0 |
| Burgers | $10^{-6}$ | regression | 2.25e1 | 2.92e1 | 2.5e1 |
| Burgers | $10^{-5}$ | interpolation | 1.34e0 | 9.17e−1 | 1.14e0 |
| Burgers | $10^{-5}$ | regression | 2.25e1 | 2.92e1 | 2.5e1 |
| Burgers | $10^{-4}$ | interpolation | 1.34e0 | 9.17e−1 | 1.14e0 |
| Burgers | $10^{-4}$ | regression | 2.25e1 | 2.92e1 | 2.5e1 |
| Burgers | $10^{-3}$ | interpolation | 1.34e0 | 9.17e−1 | 1.14e0 |
| Burgers | $10^{-3}$ | regression | 2.26e1 | 2.94e1 | 2.51e1 |
| Burgers | $10^{-2}$ | interpolation | 1.31e0 | 8.91e−1 | 1.12e0 |
| Burgers | $10^{-2}$ | regression | 2.96e1 | 3.90e1 | 3.32e1 |
| Burgers | $10^{-1}$ | interpolation | 8.72e−1 | 6.82e−1 | 7.85e−1 |
| Burgers | $10^{-1}$ | regression | 1.81e1 | 5.42e1 | 3.74e1 |
| Burgers | 1 | interpolation | 3.67e−1 | 6.48e−1 | 5.02e−1 |
| Burgers | 1 | regression | 3.31e−1 | 5.53e−1 | 4.33e−1 |
| Burgers | 10 | interpolation | 3.52e−1 | 4.92e−1 | 4.12e−1 |
| Burgers | 10 | regression | 3.34e−1 | 4.53e−1 | 3.83e−1 |
| Cavity | 0 | interpolation | failed | failed | failed |
| Cavity | 0 | regression | failed | failed | failed |
| Cavity | $10^{-6}$ | interpolation | failed | failed | failed |
| Cavity | $10^{-6}$ | regression | failed | failed | failed |
| Cavity | $10^{-5}$ | interpolation | 9.98e3 | 1.09e4 | 1.05e4 |
| Cavity | $10^{-5}$ | regression | 2.14e−1 | 2.83e−1 | 2.5e−1 |
| Cavity | $10^{-4}$ | interpolation | 9.98e3 | 1.09e4 | 1.05e4 |
| Cavity | $10^{-4}$ | regression | 2.14e−1 | 2.83e−1 | 2.5e−1 |
| Cavity | $10^{-3}$ | interpolation | 9.98e3 | 1.07e4 | 1.03e4 |
| Cavity | $10^{-3}$ | regression | 2.14e−1 | 2.82e−1 | 2.5e−1 |

*Continued on next page*

| Equation | $\lambda$ | Method | Error $[0, T]$ | Error $[T, 2T]$ | Error $[0, 2T]$ |
|---|---|---|---|---|---|
| Cavity | $10^{-2}$ | interpolation | 1.01e4 | 1.2e4 | 1.11e4 |
| Cavity | $10^{-2}$ | regression | 1.95e−1 | 2.56e−1 | 2.27e−1 |
| Cavity | $10^{-1}$ | interpolation | 4.29e−2 | 3.90e−2 | 4.11e−2 |
| Cavity | $10^{-1}$ | regression | 9.56e3 | 1.03e4 | 1.08e4 |
| Cavity | 1 | interpolation | 4.01e−2 | 3.88e−2 | 3.95e−2 |
| Cavity | 1 | regression | 8.35e−2 | 9.60e−2 | 9.00e−2 |
| Cavity | 10 | interpolation | 2.21e−1 | 1.51e−1 | 1.96e−1 |
| Cavity | 10 | regression | 2.37e−1 | 1.66e−1 | 2.08e−1 |

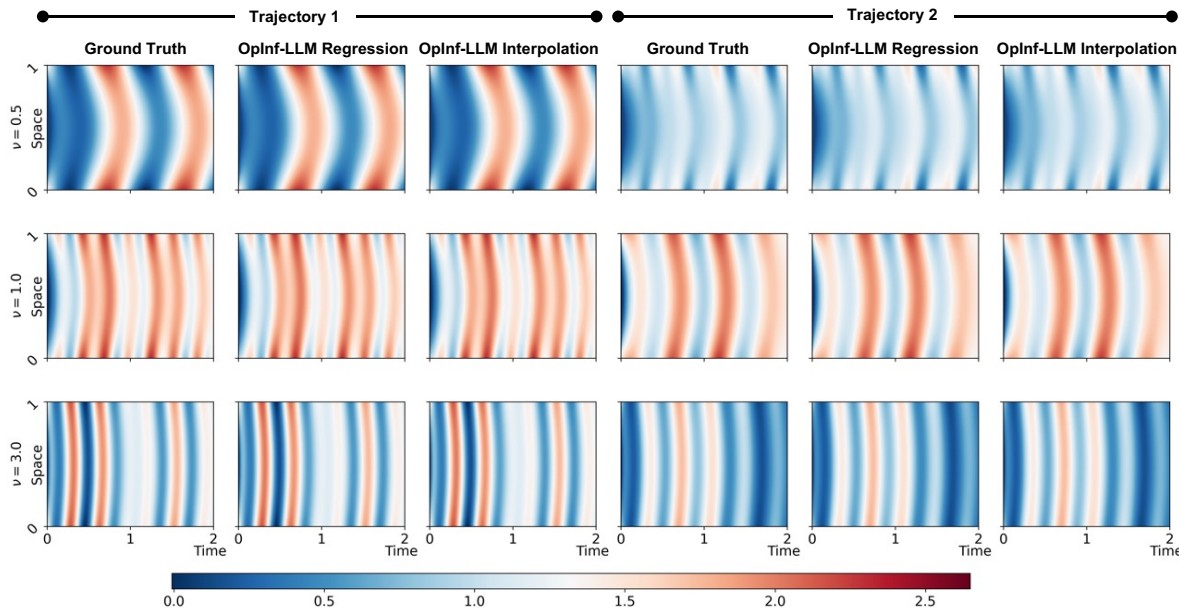

Figure 10: Heat equation predictions of different $\nu$.

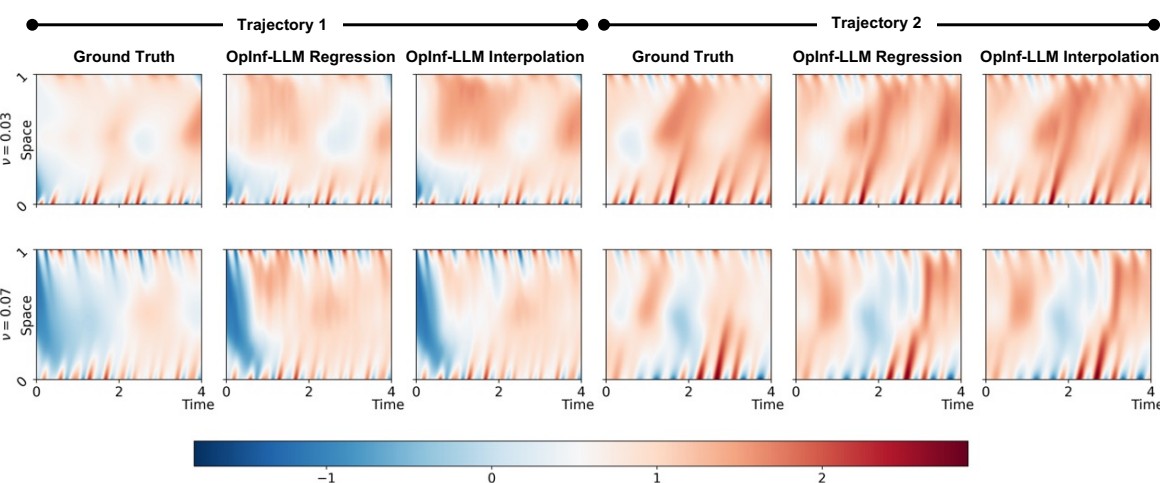

Figure 11: Burgers' equation predictions of different $\nu$.

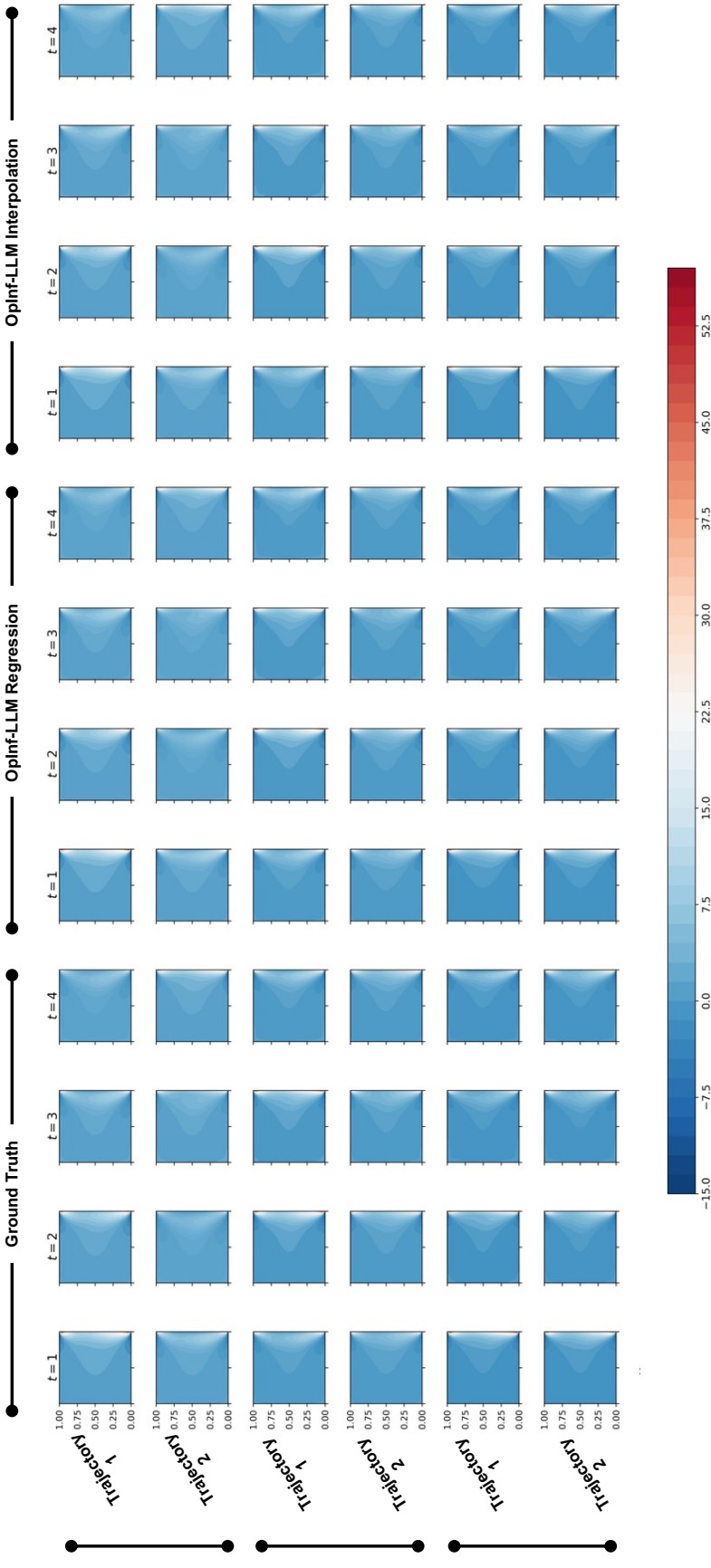

Figure 12: Cavity results across Reynolds numbers.

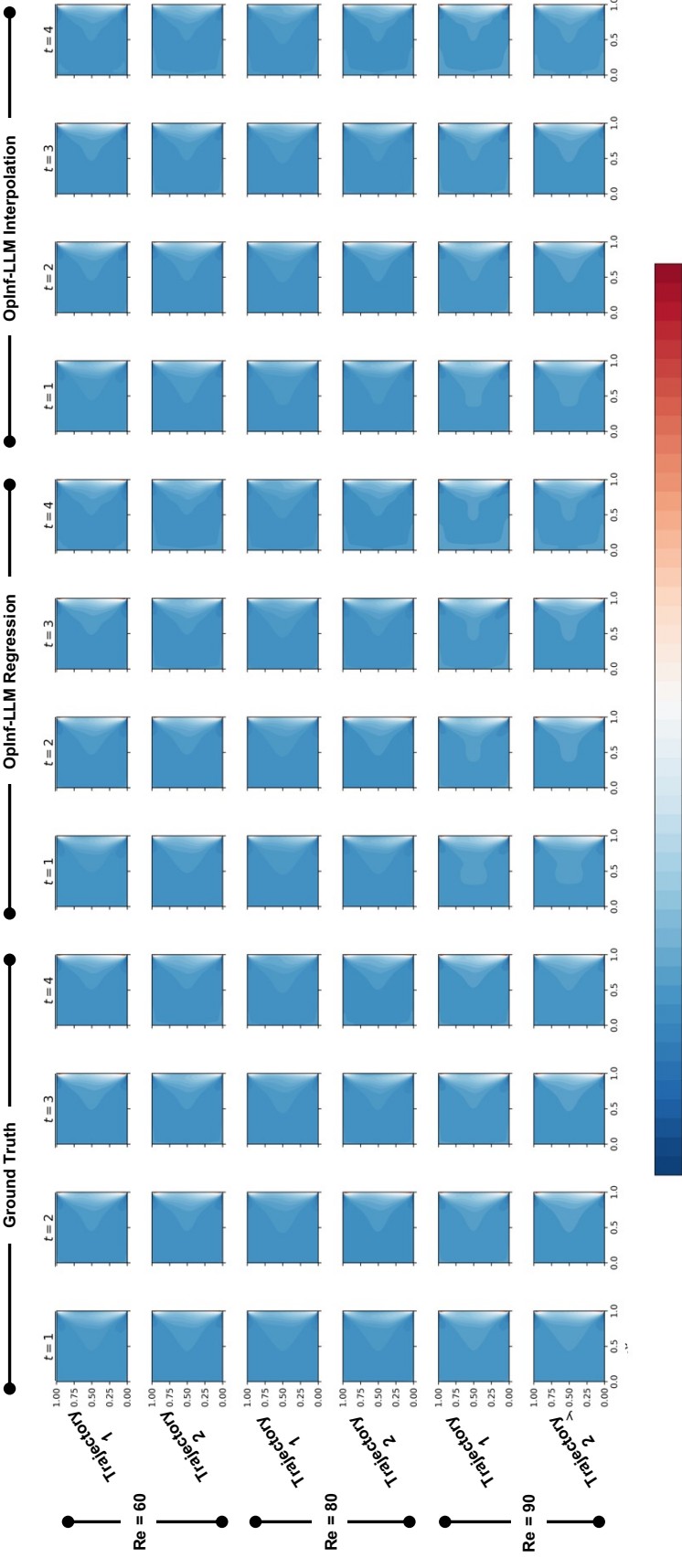

Figure 13: Cavity results across Reynolds numbers.

