# OpenReview forum: "OpInf-LLM: Parametric PDE Solving with LLMs via Operator Inference"
_TMLR — Under review for TMLR_

### Review · Reviewer_dYSK · 2026-06-16

**Summary Of Contributions:**

This paper proposes OpInf-LLM, a framework for solving parametric partial differential equations (PDEs). The framework combines operator-inference-based reduced-order modeling with LLM-based natural language parsing, tool calling, and/or code generation. The main experimental results suggest that, compared with direct LLM code generation, direct LLM solution prediction, and a multimodal PDE foundation-model baseline, the proposed design achieves a favorable balance between execution success rate and numerical accuracy.

The main strengths of the paper are:

- It uses LLMs for high-level parsing and orchestration, while relying on reduced-order numerical models for actual prediction. This is more reasonable than directly asking LLMs to solve PDEs.
- The method has relatively low computational cost at test time.
- The experimental results are encouraging.

**Audience:**

Yes

**Audience Explanation:**

The paper addresses a topic that is likely to interest parts of the TMLR audience: how to combine LLMs with scientific machine learning and numerical solvers in a way that is more reliable than direct code generation or direct black-box prediction. The idea of using LLMs as an interface and orchestration layer for reduced-order modeling is timely and practically relevant.

**Broader Impact Concerns:**

I do not see major ethical concerns that would require rejection.

**Claims And Evidence:**

Yes

**Claims Explanation:**

This paper provides useful and generally credible evidence that the proposed reduction workflow outperforms the selected baseline methods on the three PDE settings considered.

**Requested Changes:**

1. Add a non-LLM parametric OpInf baseline.

2. Clarify the fairness of baseline comparisons. The comparison to CodePDE should explicitly discuss the fact that CodePDE is asked to generate a full solver, while OpInf-LLM uses offline solution data and learned reduced operators.

3. The paper should compare against a conventional parametric ROM/OpInf baseline.

4. Clarify data counts and training time.

5.The authors should report error distributions across parameters and trajectories, not only averages.

6.The paper uses (y), (u), (c), and (C) somewhat inconsistently across the main text, appendix, and prompts. Cleaning this up would improve readability.

---

> ### Author Response · Authors · 2026-07-12
> **Author Response**
>
> We thank the reviewer for the thoughtful suggestions and constructive feedback. Please see below our response to your comments.
>
> 1 & 3. Add a non-LLM parametric OpInf baseline.
>
> We thank the reviewer for the helpful suggestion. In the revised manuscript, we added a non-LLM standard OpInf comparison in Appendix C.5 and Table 9 to isolate the role of the LLM component. In this comparison, the OpInf solvers are manually specified and implemented for each equation family, without natural-language input or LLM assistance. We include this comparison as a reference rather than in Table 1, since standard OpInf requires manual solver design and implementation, while OpInf-LLM automatically constructs and implements the solver from natural-language problem descriptions.
> The results in Table 9 show that OpInf-LLM with tool use exactly recovers the standard OpInf prediction errors and success rate. This indicates that, given the appropriate tools, the LLM can consistently construct the intended OpInf solvers. The code-generation variants achieve comparable prediction errors, with differences due to generated implementation details and feature selection.
>
>
> 2. Clarify the fairness of baseline comparisons.
>
> We thank the reviewer for this helpful comment. We have revised the paper to clarify that the baselines differ substantially in their data requirements and task setup. CodePDE and direct LLM prediction are zero-training-data methods that rely only on the problem description at inference time, while MOL-LLM is a supervised foundation-model baseline that typically requires a large amount of trajectory data for training. OpInf-LLM fills the middle ground by using a small number of offline trajectories to learn reduced operators while still accepting natural-language problem descriptions. We also clarify that all errors are evaluated after reconstruction to the full physical solution field, so the methods are compared in the same output space.
>
>
> 4. Clarify data counts and training time.
>
> We thank the reviewer for the helpful suggestion. We have revised the experiment section to clarify both the data counts and training-time definition. Across the three benchmark problems, the composite training set contains 449 trajectories in total, consisting of 9 heat-equation trajectories, 400 Burgers' trajectories, and 40 lid-driven cavity trajectories. The composite test set contains 112 trajectories in total, consisting of 60 heat-equation trajectories, 40 Burgers' trajectories, and 12 cavity-flow trajectories. We also specify the hardware used for all experiments and clarify that OpInf-LLM training time denotes the offline computation time for POD basis construction and solution projection.
>
>
> 5. Report error distributions.
>
> We thank the reviewer for the helpful suggestion. We have added trajectory-level error distribution results in Appendix C.6. The new figures report relative $L^2$ error distributions for the heat, Burgers, and lid-driven cavity test sets, where each point corresponds to one test trajectory under a specific test parameter setting. This provides a more detailed view beyond average errors.
>
>
> 6. Clean up inconsistent use of C and c.
>
> We thank the reviewer for pointing this out. We have revised the manuscript to improve notation consistency. In particular, we keep $y$ for the PDE solution and $u$ for the boundary control, and we now use lowercase $c$ consistently for the constant term in the OpInf ROM instead of mixing $c$ and $C$.

---

### Review · Reviewer_bBbE · 2026-06-24

**Summary Of Contributions:**

The paper proposes OpInf-LLM, a framework that combines operator inference based reduced-order modeling with LLM-based natural language parsing and tool/code invocation for parametric PDE solving. The method constructs reduced-order operators offline using POD and OpInf, and uses LLMs online to parse PDE specifications, identify parameters, perform operator interpolation/regression, and integrate the reduced ODE system.

The paper evaluates the method on heat, Burgers’, and lid-driven cavity equations with unseen parameters and varying boundary conditions. The main strength is a practical and robust system that improves the trade-off between execution success rate and prediction accuracy. The main weakness is that most numerical performance appears to come from the OpInf reduced-order model, while the unique contribution of the LLM component is not fully isolated.

**Additional Comments:**

In my view, this is a useful LLM-assisted OpInf system rather than evidence that LLMs themselves can directly solve PDEs. With clearer positioning and stronger ablations separating the LLM and OpInf components, the paper would be more convincing.

Overall recommendation: Borderline / Weak Accept.

**Audience:**

Yes

**Audience Explanation:**

The paper is relevant to TMLR audiences interested in scientific machine learning, neural PDE solvers, reduced-order modeling, and LLM-assisted scientific computing. It provides a practical alternative to direct LLM code generation and large PDE foundation models by combining classical reduced-order modeling with language interfaces.

The work may be especially useful for researchers interested in building more reliable scientific AI systems, even if the methodological novelty is more at the system-integration level than at the level of a new PDE solver.

**Broader Impact Concerns:**

I do not identify major ethical concerns specific to this work. The method is intended for scientific computing and PDE simulation. The main practical concern is that automated solvers should be carefully validated before use in safety-critical engineering or scientific applications.

**Claims And Evidence:**

Yes

**Claims Explanation:**

The experiments generally support the claim that the overall framework is accurate and robust on the tested PDE families. The paper provides comparisons with CodePDE, MOL-LLM, and direct LLM prediction, as well as ablations on temporal extrapolation, natural language parsing, POD modes, and regularization.

However, the evidence is less convincing for stronger claims about LLM reasoning. The LLM mainly acts as a parser and controller for OpInf tools, while the core PDE-solving accuracy comes from the reduced-order model. The paper should therefore more clearly separate the contribution of OpInf from the contribution of the LLM.

**Requested Changes:**

Critical:
1. Clarify the role of the LLM component. The paper should explicitly state that the LLM mainly performs parsing, tool invocation, and workflow orchestration, while the numerical prediction is primarily handled by OpInf.
2. Add or discuss baselines that isolate the LLM contribution, such as OpInf with structured input, OpInf with a rule-based parser, or OpInf without natural language input.
3. Better position the novelty relative to existing parametric OpInf. Shared POD bases, reduced operators, and parameter interpolation/regression are standard components of OpInf-based ROM pipelines, so the paper should clarify what is new beyond the LLM interface.
4. Moderate the claims about physical reasoning. The 1/Re example is reasonable but relatively simple, since this parameterization is already explicit in the cavity-flow equation.

Non-critical but useful:
5. Discuss experimental fairness when comparing with neural PDE foundation models, since OpInf works in a reduced-order space while neural baselines often learn the full solution operator.
6. Provide more discussion of limitations, especially for new PDE families, non-quadratic dynamics, poor reduced-basis representation, and far out-of-distribution parameters.

---

> ### Author Response · Authors · 2026-07-12
> **Author Response**
>
> We thank the reviewer for the thoughtful suggestions and constructive feedback. Please see below our response to your comments.
>
>
> 1. Clarify the role of the LLM component.
>
> We have revised the related-work discussion to clarify the division of roles between the LLM and OpInf components. Specifically, in our framework, the LLM uses its reasoning and coding capabilities to interpret the natural-language PDE description, determine the appropriate reduced model structure and features, and automatically construct an executable OpInf-based solver. We emphasize that standard OpInf workflows require the model structure, regression features, data-processing steps, and solver implementation to be manually specified and coded for each PDE family, whereas our framework automates these steps from natural-language problem descriptions across diverse problem families.
>
>
> 2 & 3. Discuss baselines that isolate the LLM contribution and better position the novelty relative to existing parametric OpInf.
>
> In the revised manuscript, we added Appendix C.5 to compare OpInf-LLM with standard OpInf solvers that are manually specified and implemented for each equation family, without natural-language descriptions or LLM assistance. This comparison is included as a reference rather than a direct baseline, since standard parametric OpInf does not take natural-language instructions as input and requires manual configuration of the equation family, operator structure, regression features, data-processing steps, and solver implementation.
>
> We also revised the related-work discussion and Appendix C.5 to clarify the novelty of OpInf-LLM relative to existing parametric OpInf. We agree that shared POD bases, reduced operators, and parameter-dependent regression are standard OpInf components, and we do not claim these as new. Rather, OpInf-LLM uses these components as a structured framework for language-guided solver generation. The results in Table 9 show that OpInf-LLM with tool use exactly recovers the standard OpInf prediction errors, indicating that the LLM can consistently construct the intended OpInf solvers from natural-language problem descriptions when provided with the appropriate tools. The code-generation variants also achieve comparable prediction errors, with small variations due to differences in generated implementation details and feature selection.
>
>
> 4. Moderate the claims about physical reasoning.
>
> We have moderated the claim in the experiment section and now describe the test as evaluating whether OpInf-LLM can identify operator features consistent with the governing equation, rather than as broad physical reasoning.
> We also added a feature-reasoning study in Appendix C.4 using an advection-diffusion equation on a parameter-dependent physical domain, where the parameter does not appear explicitly as a PDE coefficient. After mapping to a fixed reference domain, the correct feature map reveals inverse dependence on the domain length for advection and inverse squared dependence for diffusion. Given only the physical-domain problem statement, OpInf-LLM correctly identifies these features, matches the oracle feature map, and outperforms a raw-parameter OpInf model.
>
> The resulting relative errors are:
>
> | Test parameter | Raw-parameter ROM | OpInf-LLM |
> |:---:|:---:|:---:|
> | 0.75 | 3.0119e-2 | 4.5243e-4 |
> | 1.60 | 7.7841e-2 | 5.2922e-4 |
> | 2.00 | 2.8715e+0 | 5.8492e-4 |
>
> This additional test provides a more meaningful evaluation while keeping the claim appropriately scoped.
>
>
>
> 5. Discuss experimental fairness when comparing with neural PDE foundation models.
>
> We agree that OpInf-LLM and neural PDE foundation models use different internal representations. OpInf-LLM learns dynamics in reduced coordinates, while neural baselines typically learn solution operators on the full spatial discretization. We have clarified that OpInf-LLM predictions are reconstructed to the full physical solution field before evaluation, so all reported errors are computed in the same output space. We also emphasize that the comparison should be viewed as an end-to-end comparison of PDE-solving workflows, rather than an architecture-matched comparison.
>
>
> 6. Provide more discussion of limitations.
>
> We have expanded the limitations discussion in the revised manuscript to clarify that new PDE families usually require constructing new reduced bases offline, and that performance depends on the quality of the reduced representation. We now discuss potential failure modes for transport- or convection-dominated problems, sharp gradients, far out-of-distribution parameters or boundary conditions, and cases where a low-dimensional linear POD basis or standard OpInf structure is insufficient. We also added discussion on possible extensions, including nonlinear-manifold OpInf formulations and robustness-enhanced OpInf variants, which could be potentially incorporated into the same language-guided workflow.

---

### Review · Reviewer_myQf · 2026-07-01

**Summary Of Contributions:**

The paper proposes OpInf-LLM, a two-stage framework for parametric PDE solving. Offline, it builds a shared POD basis and fits reduced-order operators (A, H, B, c) via least squares for a finite set of training parameters. Online, an LLM parses a natural-language task description, then interpolates or regresses those operators to an unseen parameter and integrates the resulting low-dimensional ODE. It is evaluated on the 1D heat equation, forced viscous Burgers, and 2D lid-driven cavity, against CodePDE (LLM solver generation), MOL-LLM (multimodal PDE foundation model), and direct LLM prediction, across GPT-4.1, GPT-4o, Gemini-2.0-flash, and (in the appendix) Qwen Plus and Claude Sonnet 4.

Strengths: the OpInf-plus-LLM combination is, per the authors' related-work survey, not previously done; the framework gets a high execution success rate (99.2%) with far lower training cost than MOL-LLM (30s vs 14306s), and the NL-parsing robustness result (Table 4: 100% vs 52% exact match) is clean and convincing.

Weaknesses: the accuracy claim is uneven (Burgers relative L2 stays near 0.49); the LLM backend has almost no effect on the numerical results, which complicates the "LLM integration" framing; and the baseline comparison mixes methods that do and do not have access to solution data.

**Audience:**

Yes

**Audience Explanation:**

The scientific-ML, reduced-order-modeling, and LLM-for-science segments of TMLR's readership would find the bridging of operator inference with LLM-driven task specification worth knowing, along with the concrete result that a lightweight ROM pipeline can match or beat a trained PDE foundation model at a fraction of the training cost while accepting natural-language input.

**Broader Impact Concerns:**

No broader impact statement is present. Given the domain (numerical PDE solving on standard benchmarks with no personal data, no obvious dual-use, and no generative-content risk), the absence is acceptable.

I do not see ethical implications requiring a mandatory statement.

**Claims And Evidence:**

No

**Claims Explanation:**

1. Accuracy "across all equations." On Burgers, OpInf-LLM's error is 4.91e-1, and CodePDE (GPT-4.1) reaches 1.50e-1 on the same equation (Table 1). So OpInf-LLM is not the most accurate method on Burgers even before accounting for its data advantage. A ~49% relative L2 error described as "accurate prediction" needs qualification.
2. "Seamless integration with LLMs" as a contribution. In tool mode the heat/Burgers/cavity numbers are identical across GPT-4.1, GPT-4o, and Gemini (Table 1), and in Table 8 they are identical across five backends including Claude Sonnet 4 and Qwen, with only minor cavity differences. This is consistent with the LLM doing parsing and orchestration while a deterministic tool produces the numbers. That is a reasonable design, but it means the LLM choice does not affect accuracy, so the framework's numerical performance is really that of parametric OpInf; the evidence does not show the LLM contributing to accuracy beyond parsing and the single feature-selection ablation.
3. Baseline fairness. OpInf-LLM and MOL-LLM use 449 training trajectories; CodePDE and direct LLM receive only equation descriptions (Data column "–"). Comparing success rate and error across these on equal footing overstates the gap; the data asymmetry should be stated where the comparison is drawn, not left to be inferred from the table.
4. The physical-reasoning claim rests on one equation (cavity, 1/Re) with a prompt that already tells the model to "choose a feature that yields smoother, better-conditioned operator trends," which points toward the answer. That is one data point, not evidence of general physics reasoning.

**Requested Changes:**

Add a non-LLM parametric OpInf baseline to Table 1 with success rate and error, so the LLM's marginal value over plain OpInf is isolated. Table 3 only varies the feature, not the presence of the LLM.

Explain why tool-mode success is 99.2% rather than 100% given a deterministic interpolation tool, and identify what fails. Table 8 shows Burgers ν=0.03 at 95%; reconcile this with the headline number.

Address the Burgers accuracy directly: state that ~0.49 relative L2 is high, whether it is acceptable for the intended use, and why OpInf-LLM trails CodePDE's best solver there.

State the training-data asymmetry between OpInf-LLM/MOL-LLM and CodePDE/direct-LLM at the point of comparison.

Substantiate or soften the physics-reasoning claim: use a neutral prompt that does not signal the target feature, and test on more than one equation.

Explain the near-identical results across LLM backends and what it implies for the integration contribution.

Make Fig 1 quantitative or clearly label it as schematic, with defined axes.

---

> ### Author Response · Authors · 2026-07-12
> **Author Response**
>
> We thank the reviewer for the thoughtful suggestions and constructive feedback. Please see below our response to your comments.
>
> 1. Add a non-LLM parametric OpInf baseline.
>
> We thank the reviewer for the helpful suggestion. In the revised manuscript, we added a non-LLM standard OpInf comparison in Appendix C.5 and Table 9 to isolate the role of the LLM component. In this comparison, the OpInf solvers are manually specified and implemented for each equation family, without natural-language input or LLM assistance. We include this comparison as a reference rather than in Table 1, since standard OpInf requires manual solver design and implementation, while OpInf-LLM automatically constructs and implements the solver from natural-language problem descriptions.
> The results in Table 9 show that OpInf-LLM with tool use exactly recovers the standard OpInf prediction errors and success rate. This indicates that, given the appropriate tools, the LLM can consistently construct the intended OpInf solvers. The code-generation variants achieve comparable prediction errors, with differences due to generated implementation details and feature selection.
>
>
> 2. Explain error calculation.
>
> We thank the reviewer for pointing out this ambiguity. We have revised the experiment section to clarify that the 99.2% success rate in Table 1 is the overall code execution success rate computed over all testing trajectories, while the 95% success rate in Table 8 is specific to the Burgers case with $\nu=0.03$. The only failure comes from the Burgers test, where one boundary condition leads to an unstable reduced-order model. This setting is challenging due to its nonlinear advective dynamics and sensitivity to boundary conditions, and we have added this discussion of the failure cause in the revised manuscript.
>
>
> 3. Address the Burgers accuracy.
>
> We thank the reviewer for this helpful comment. We have revised the results discussion to state more directly that the Burgers relative $L^2$ error is high and should be viewed as a challenging case rather than a fully satisfactory high-accuracy result. Burgers is difficult due to its nonlinear advective dynamics and sensitivity to boundary conditions, and it is also the only setting where OpInf-LLM has an unstable-ROM failure case.
>
>
> 4. State the training-data asymmetry.
>
> We thank the reviewer for this helpful suggestion. We have revised the experiment section to explicitly state the training data asymmetry. CodePDE and direct LLM prediction use zero training data, MOL-LLM requires substantially more trajectory data, and OpInf-LLM lies between these regimes by using a small number of trajectories to learn a reduced-order model. We also clarify that all errors are evaluated on the reconstructed full physical solution field, so the methods are compared in the same output space.
>
>
> 5. Soften the physics-reasoning claim.
>
> We thank the reviewer for this helpful suggestion. We have softened the claim in the experiment section and now describe the test as evaluating whether OpInf-LLM can identify operator features consistent with the governing equation, rather than making a broad physics-reasoning claim. We also added a new feature-reasoning study in Appendix C.4 using a neutral prompt that provides only the physical domain problem statement and does not mention the target feature map. In this additional advection-diffusion example, the parameter does not appear explicitly as a PDE coefficient, but mapping to a fixed reference domain reveals different parameter scalings for the advection and diffusion operators. OpInf-LLM correctly identifies these features and substantially outperforms a standard OpInf model.
>
>
> 6. Explain the near-identical results for OpInf-LLM with tool use.
>
> We thank the reviewer for this helpful comment. We have revised the manuscript and added a comparison with standard parametric OpInf in Appendix C.5. The near-identical tool-use results across LLM backends are expected and can be viewed as an advantage, since they show that OpInf-LLM can construct the intended solvers consistently despite using different LLM backends, while still taking natural-language problem descriptions as input and automating the solver construction process compared with standard OpInf.
>
> 7. Elaborate on the axis definitions in Fig. 1.
>
> We thank the reviewer for this helpful suggestion. Fig. 1 is intended to be quantitative rather than purely schematic. We have revised the caption to make this explicit and to define the plotted quantities. The figure now states that it shows the trade-off between prediction error and solver execution success rate, with the mean and variance computed from the experimental results in Section 4 across the three benchmark problems considered in the study.